# Improving modelled albedo over the Greenland ice sheet through parameter optimisation and MODIS snow albedo retrievals

Nina Raoult[1], Sylvie Charbit[1], Christophe Dumas[1], Fabienne Maignan[1], Catherine Ottlé[1], and Vladislav Bastrikov[2]

[1]Laboratoire des Sciences du Climat et de l'Environnement, LSCE/IPSL, CEA-CNRS-UVSQ, Université Paris-Saclay, Gif-sur-Yvette, France
[2]Science Partners, Paris, France

**Correspondence:** Nina Raoult (nina.raoult@lsce.ipsl.fr)

**Abstract.** Greenland ice sheet mass loss continues to accelerate as global temperatures increase. The surface albedo of the ice sheet determines the amount of absorbed solar energy, which is a key factor in driving surface snow and ice melting. Satellite retrieved snow albedo allows us to compare and optimise modelled albedo over the entirety of the ice sheet. We optimise the parameters of the albedo scheme in the ORCHIDEE land surface model for three random years taken over the 2000-2017 period and validate over the remaining years. In particular, we want to improve the albedo at the edges of the ice sheet since they correspond to ablation areas and show the greatest variations in runoff and surface mass balance. By giving a larger weight to points at the ice sheet's edge, we improve the model-data fit by reducing the root-mean-square deviation by over 25% for the whole ice sheet for the summer months. This improvement is consistent for all years, even those not used in the calibration step. We also show the optimisation successfully improves the model-data fit at 87.5% of in situ sites from the PROMICE network. We conclude by showing which additional model outputs are impacted by changes to the albedo parameters encouraging future work using multiple data streams when optimising these parameters.

## 1 Introduction

The melting of the Greenland ice sheet (GrIS) is one of the main contributors to sea-level rise (Frederikse et al., 2020). As global temperatures continue to increase under climate change, further melting and surface mass loss are expected (The IMBIE team, 2020), potentially affecting deep ocean circulation (Hu et al., 2011). Increased warming also darkens the GrIS (Tedesco et al., 2016), decreasing the surface reflectivity (i.e. albedo). This darkening has already been observed over the last decades, driven by snowmelt, the retreat of the snow line, dust deposition (Dumont et al., 2014), and algae growth (Perini et al., 2019; Cook et al., 2020; Williamson et al., 2020) and is expected to worsen. Since surface albedo determines the land surface energy balance by controlling the amount of reflected solar (shortwave) radiation, reductions in albedo - through the darkening of the ice sheet - result in increased shortwave absorption. This, in turn, enhances melting, creating a strong feedback to the atmosphere (Charbit et al., 2019; Box et al., 2022). The melt-albedo feedback is an essential contributor to mass loss (Qu and Hall, 2014; Zeitz et al., 2021) and can be used as an emergent constraint to reduce the inter-model variability in projections of climate change (Thackeray et al., 2021).

Both dynamical effects and surface processes drive Greenland's evolution. However, recent studies show that surface mass balance changes dominate mass balance changes (Van den Broeke et al., 2016; Ryan et al., 2019; The IMBIE team, 2020). To correctly represent the surface mass balance and its components (sublimation and runoff), it is important to simulate the physical processes within the snowpack. These depend on the surface energy balance and, therefore, on the albedo. Given the importance of this albedo, it is crucial that it is accurately simulated in land surface models (LSMs) used to generate climate change projections. Therefore it is important to confront LSM albedo estimates with observed values. With large areas such as the GrIS, we can rely on remote sensing-based albedo measurements derived from various polar-orbiting satellites (Qu et al., 2015). We can use these data to evaluate and optimise LSMs using data assimilation.

Data assimilation (DA) refers to the act of incorporating observational information into a model to constrain its estimates or parameters. Several studies have used remotely sensed albedo for DA in LSMs. Due to albedo's influence on the partitioning of the surface energy fluxes and the subsequent effect on the development of planetary boundary conditions and clouds (Pielke and Avissar, 1990), some studies have focused on the impact assimilating surface albedo has on numerical weather prediction (e.g., Cedilnik et al., 2012; Boussetta et al., 2015). Others have mainly used remotely sensed data to derive new vegetation and soil background albedo parameters to use in land surface models (e.g., Liang et al., 2005; Houldcroft et al., 2009). There are also a number of examples of using snow albedo to improve snow models. For example, Malik et al. (2012) used MODIS (Moderate Resolution Imaging Spectroradiometer; Schaaf et al. (2002))-based snow albedo and direct insertion methodology in the Noah LSM over three sites in Colorado to improve simulated snow depth and snow season duration. Satellite-based albedo data was also used by Wang et al. (2015) to calibrate the ORCHIDEE (ORganizing Carbon and Hydrology in Dynamic Ecosystems, Krinner et al. (2005)) LSM and investigate the impacts of albedo assimilation on offline and coupled model simulations. Dumont et al. (2012) assimilated remotely sensed albedo in the Crocus snowpack model (Vionnet et al., 2012) to improve the modelling of the spatial distribution of the glacier mass balance. Navari et al. (2018) further improved the Crocus model using satellite-derived albedo to improve surface mass balance (SMB) along Greenland's Kangerlussuaq transect. Other datasets have also been assimilated to improve snow estimation, including snow cover fraction estimates from optical sensors (e.g., Toure et al., 2018; Xue et al., 2019) and measured ice surface temperatures (e.g., Navari et al., 2018). There have also been several studies assimilating joint datasets. For example, MODIS-based snow cover fraction and albedo have been assimilated in the Common Land Model LSM (Xu and Shu, 2014) and the Noah LSM (Kumar et al., 2020). All these snow model studies use DA for state estimation, i.e., updating the model state whilst keeping the model parameters fixed. The techniques used range from relatively simple methods like direct insertion to more advanced statistical techniques like the ensemble Kalman filter and particle filters.

Examples of DA used for parameter estimation, i.e., optimising internal model parameters, in snow modelling are less common. Su et al. (2011) demonstrated how DA can be used for joint state and parameter assimilation in snow modelling. Nevertheless, DA for parameter estimation remains more commonly used by the LSM community to optimise vegetation parameters (see orchidas.ipsl.lsce.fr for such examples calibrating the ORCHIDEE LSM). In these types of studies, it is common to optimise over a single site (or single pixel) or a group of individual pixels, usually sharing a common trait (e.g. the dominant vegetation present), in what is known as a "multisite" approach (e.g., Kuppel et al., 2012; Raoult et al., 2016). In each case,

the optimisation results in sets of parameters that apply to that individual site or trait tested. These approaches were used because, historically, models were optimised against in situ measurements from sites that are sparsely and unevenly distributed. Advances in satellite data retrieval have helped provide data over large areas for which we previously had no measurements. However, with large amounts of data, computational power and time still limit the experiments we can perform.

Using MODIS snow albedo, in this study, we use DA for parameter estimation to improve the albedo parameterisation inside the ORCHIDEE LSM (Krinner et al., 2005). While albedo parameters in ORCHIDEE have been optimised for vegetation and bare soil, this will be the first study optimising them for ice sheets. Our target area from this study is the Greenland Ice Sheet. This study is the first test of applying the ORCHIDEE data assimilation system over ice sheets to improve modelling albedo and, in turn, the surface mass balance of the ice sheet. Instead of using a single or multisite approach which samples the space, here, to exploit the full spatial coverage of the satellite retrievals, we optimise over the whole area of the GrIS to obtain one best set of model parameters applicable over the full ice sheet. Although this study is only over the GrIS, we can apply the method to other regions. We show how robust Bayesian parameter estimation is an important tool for model development. We further highlight the different limitations and considerations needed to apply such an approach. The paper is organised as follows. Methods and data, including the details about the ORCHIDEE LSM and its DA framework, driving and observational datasets, and performed experiments, can be found in Sect. 2. Section 3 lists the results, starting with an assessment of the prior. This is followed by the results of the main experiments and an evaluation over PROMICE in situ sites. In Sect. 3.3, we look at the impact of the optimisation on the modelling of the SMB of the GrIS, as well as the different SMB components. In this section, we also perform a sensitivity analysis of the different parameters of the snow model for future work. Finally, the discussion and conclusions can be found in Sect. 4.

## 2 Methods and Data

### 2.1 ORCHIDEE land surface model

The ORCHIDEE land surface model (Krinner et al., 2005) is the terrestrial component of the IPSL Earth System Model (ESM) used in climate projections (Boucher et al., 2020; Cheruy et al., 2020). Either run off-line (i.e., driven by prescribed meteorological forcing) or coupled with an atmospheric model (i.e., as part of the ESM), ORCHIDEE describes the exchanges of energy, water, and carbon between the atmosphere and the continental biosphere. The land surfaces are represented as fractions of bare soil and plant functional types. These surfaces can further be covered with snow.

In this study, we adapted the CMIP (Coupled Model Intercomparison Project) 6 version of ORCHIDEE to run over the GrIS. The CMIP6 version of ORCHIDEE uses the three-layered snow model presented in Wang et al. (2013). To apply ORCHIDEE over the GrIS, we implemented a new soil type into this version of ORCHIDEE to mimic the presence of ice in regions defined by the present-day ice mask (Bamber et al., 2013). In ORCHIDEE, each soil type is defined according to the USDA (United States Department of Agriculture) taxonomy, which classifies soils as a function of their chemical, physical and biological properties (Carsel and Parrish, 1988). For the new icy soil type, the porosity and the saturated volumetric water content are set to 0.98 to simulate a soil filled with frozen water. This amounts to considering that ice is an impermeable medium. However,

it does not allow the representation of processes such as moulins where water seeps through a network of galleries because the model does not simulate the lateral transport of water. All the other characteristics of this new soil type were set to those of the loam soil type because it is the dominant soil type in the ice-free regions around the GrIS (Fischer et al., 2008). Furthermore, to be able to compare directly modelled to satellite retrieved albedo values, we computed the mean of albedo in both visible (VIS) and near-infrared (NIR) spectral domains. This is done to be in accordance with MODIS data. We only consider this averaged albedo in the rest of the study.

The snow albedo in ORCHIDEE is modelled following the formulation of Chalita and Le Treut (1994). In the absence of fresh snow, snow-covered albedo in ORCHIDEE ($\alpha_{snow}$) decreases exponentially with time from its fresh value ($A_{aged}+B_{dec}$) to a minimum value after ageing, i.e. albedo of old snow ($A_{aged}$),

$$\alpha_{snow} = \mathbf{A_{aged}} + \mathbf{B_{dec}} \exp\left(-\frac{\tau_{snow}}{\boldsymbol{\tau_{dec}}}\right). \tag{1}$$

Here the $B_{dec}$ and $\tau_{dec}$ parameters control the decay rate of snow albedo. This formula can be used to calculate the snow-covered albedo over different vegetation types, with different values of $A_{aged}$ and $B_{dec}$ accounting for the variability of snow coverings. The parameterisation of snow age, $\tau_{snow}$, is shown in Eq. 2,

$$\tau_{snow}(t+dt) = \tau_{snow}(t) + f_{age} \tag{2}$$

where $t$ is the time, $dt$ is the model time step (1800s). The latter term of equation, $f_{age}$, represents the effect of low temperatures on metamorphism,

$$f_{age} = \left[\frac{\left(\tau_{snow}(t) + \left(1 - \frac{\tau_{snow}}{\boldsymbol{\tau_{max}}}\right) \cdot dt\right) \cdot \exp\left(-\frac{P_{snow}}{\boldsymbol{\delta_c}}\right) - \tau_{snow}(t)}{1 + g_{temp}(T_{soil})}\right]; \qquad g_{temp}(T_{soil}) = \left[\frac{\max(T_0 - T_{soil}, 0)}{\boldsymbol{\omega}}\right]^{\boldsymbol{\beta}} \tag{3}$$

where $P_{snow}$ is snowfall, $\delta_c$ is the snowfall depth required to reset the age of the snow, $\tau_{max}$ is the maximum snow age, $T_0$ is the melting temperature (0°C), $T_{soil}$ is soil temperature, and $\omega$ and $\beta$ are tuning constants. All the parameters in bold are listed in Table 1. These, along with the albedo of ice, $\alpha_{ICE}$, are the parameters we focused on in this study.

## 2.2 Driving and observational datasets

### 2.2.1 Forcing provided by regional model (MAR)

The ORCHIDEE model was forced using meteorological outputs from the regional climate model Modèle Atmosphérique Régional (MAR; Gallée and Schayes (1994); Kittel (2021)), version 3.11.4. MAR is a regional atmospheric model that uses 6 hourly ERA-Interim reanalyses data from the European Centre for Medium-Range Weather Forecasts (ECMWF, Dee et al. (2011)) to prescribe the atmospheric boundary conditions outside the domain. Outputs from the MAR have a resolution of 20 km and a 3 hourly time step. In addition to the MAR meteorological outputs, we consider runoff, sublimation and SMB outputs in this study to assess the impact of the optimisation on these simulated quantities. MAR was specifically developed for polar regions and offers good performances for the calculation of SMB and its components. Furthermore, it has been shown

**Table 1.** Parameters of the snow model. The default values represent the values used in the standard simulation of ORCHIDEE, min and max refer to the range over which the parameters are allowed to vary during our experiments.

| Parameter | Description | Name in code | Default values | Min | Max |
|---|---|---|---|---|---|
| $A_{aged}$ | Albedo of old snow | SNOWA_AGED* | 0.62 | 0.50 | 0.70 |
| $B_{dec}$ | Sum with $A_{aged}$ to be the albedo of fresh snow | SNOWA_DEC* | 0.169 | 0.10 | 0.40 |
| $\delta_c$ | Snowfall depth required to reset the snow age (m) | SNOW_TRANS_NOBIO | 0.2 | 0.2 | 2 |
| $\tau_{dec}$ | Snow age decay rate (days) | TCST_SNOWA_NOBIO | 10 | 1 | 10 |
| $\omega$ | Tuning constants for glaciated snow covered areas | OMG1 | 7 | 1 | 7 |
| $\beta$ | | OMG2 | 4 | 0.5 | 4.5 |
| $\tau_{max}$ | Maximum snow age (days) | MAX_SNOW_AGE | 50 | 40 | 60 |
| $\alpha_{ICE}$ | Ice albedo | ALB_ICE | 0.4 | 0.3 | 0.5 |

* note the sum of $A_{aged}$ and $B_{dec}$ must be less than or equal to 1 - this constraint is enforced during the optimisations.

to outperform reanalysis products such as ERA5 (Delhasse et al., 2020), especially in providing the near-surface temperature in summer which play a critical role in representing snow and ice processes.

### 2.2.2 MODIS snow albedo

In this study, we used satellite-derived snow albedo from the NASA (National Aeronautics and Space Administration) MODIS
MOD10A1 product (Hall et al., 1995). This product uses data from the Terra satellite, which has a sun-synchronous, near-polar circular orbit crossing the equator at approximately 10:30 A.M. local time (Hall and Riggs, 2016) and providing global coverage every 1-2 days. MOD10A1 is a clear-sky daily product. When more than one retrieval is available on a given day, which is the case near the poles, the best value is kept. This best value is chosen based on solar elevation, distance from nadir and cell coverage (Hall and Riggs, 2016). In addition, pixels in the MOD10A1 with solar zenith angles greater than 70° are
masked (night is defined as a solar zenith angle greater than 85°). Note that this dataset does not include data from the Aqua satellite.

The version of MOD10A1 we used in this study was further processed by Box et al. (2017). Using data from collection 6 of MOD10A1 (Riggs et al., 2015; Hall and Riggs, 2016), Box et al. (2017) de-noised, gap-filled and calibrated the data into a daily 5km grid covering Greenland for the years 2000-2017. This dataset was further validated against ground-based measurements
from the PROMICE (Programme for Monitoring of the Greenland Ice Sheet) stations (Fausto et al., 2021) and the residual bias in the dataset based on the solar zenith angle corrected for using a linear regression according to time and latitude (Box et al., 2017). Finally, in this dataset, the April values are used for the winter months (January, February, November, and December). This is because there is inadequate solar illumination to compute the albedo during these months.

In this study, we used this dataset processed by Box et al. (2017), further aggregating these data using bilinear interpolation
to the resolution of the ORCHIDEE outputs, imposed by the meteorological forcing files (20 km).

### 2.2.3 PROMICE in situ data

Albedo observations from the PROMICE in situ network were used to evaluate the optimisation. The PROMICE program was initiated in 2007 (Ahlstrøm et al., 2008; van As et al., 2011), creating a network of on-ice automatic weather stations to provide in situ measurements of accumulation, ablation, and energy balance of the GrIS. Most sites come in pairs, with a lower station (L) placed near the ice sheet margin and an upper station (U) placed higher up in the ablation area (Fausto et al., 2021). As such, the majority of sites are found at the edges of the ice sheet. In some regions, there are also additional stations, for example, in the middle (M) of the lower and upper stations. The sites used in this study are listed in Table 2. We started the analysis with the year where all of March to November was available and ended the analysis with the year 2017 (or the last operational year) to be consistent with the rest of the work. Further information on ground measurements of snow albedo and associated methodology can be found in Fausto et al. (2021).

## 2.3 Data assimilation system for the ORCHIDEE LSM

### 2.3.1 A Bayesian framework

To perform the optimisations, we used ORCHIDAS, the ORCHIDEE data assimilation system. ORCHIDAS is a variational DA system in which all observations within the assimilation time window are included in the optimisation. It uses a Bayesian statistical formalism (Tarantola, 2005) where errors associated with the parameters, the observations, and the model outputs are assumed to follow Gaussian distributions. The optimal parameter set corresponds to the minimum of a cost function, J(**x**):

$$J(\mathbf{x}) = \frac{1}{2} \left[ (\mathbf{y} - M(\mathbf{x}))^T \mathbf{R}^{-1} (\mathbf{y} - M(\mathbf{x})) + (\mathbf{x} - \mathbf{x}_b)^T \mathbf{B}^{-1} (\mathbf{x} - \mathbf{x}_b) \right] \tag{4}$$

where J(**x**) measures the mismatch between (i) the observations **y** and the corresponding model outputs $M(\mathbf{x})$ (where $M$ is the model operator), and (ii) the a priori ($\mathbf{x}_b$) and optimised parameters (**x**). Each term is weighted by its error covariance matrices, **R** and **B**. As in most studies, we set both matrices to be diagonal. For the **B** matrix, we define the prior distribution of each parameter to be 40% of the prior range. For the **R** matrix, we defined the observation error (variance) as the mean-squared difference between the observations and the prior model simulation so that this variance reflects not only the measurement errors but also the model errors. Although not ideal, this approach is common since it is one of the only ways we can assess the model structural error, which is a large contributor to the **R** matrix. This error was approximately 0.06 at the edge of the ice sheet to 0.02 in the middle.

To minimise the cost function, we use a stochastic random search method, the genetic algorithm (GA), which belongs to a larger class of evolutionary algorithms that follows the principles of genetics and natural selection (Goldberg, 1989; Haupt and Haupt, 2004). With each gene corresponding to a different parameter, a vector of parameters is considered to be a chromosome. At each iteration, $p$ chromosomes are created (where $p$ is the population selected by the user, here chosen to be 30). For the first set of chromosomes, the parameters are randomly perturbed. For subsequent iterations, the chromosomes are created from the previous iteration by one of two processes. The first is the "crossover" process. This is the exchange of the gene sequences of two parent chromosomes. The second process is "mutation", where selected genes of one parent are randomly perturbed.

| Site name | Latitude (° N) | Longitude (° W) | Elevation (m a.s.l.) | Years used |
|---|---|---|---|---|
| KPC_L | 79.9108 | 24.0828 | 370 | 2009-2017 |
| KPC_U | 79.8347 | 25.1662 | 870 | 2009-2017 |
| THU_L | 76.3998 | 68.2665 | 570 | 2011-2017 |
| THU_U | 76.4197 | 68.1463 | 760 | 2011-2017 |
| EGP | 75.6247 | 35.9748 | 2660 | 2017 |
| UPE_L | 72.8932 | 54.2955 | 220 | 2010-2017 |
| UPE_U | 72.8878 | 53.5783 | 940 | 2010-2017 |
| SCO_L | 72.223 | 26.8182 | 460 | 2009-2017 |
| SCO_U | 72.3933 | 27.2333 | 970 | 2009-2017 |
| KAN_L | 67.0955 | 49.9513 | 670 | 2009-2017 |
| KAN_M | 67.067 | 48.8355 | 1270 | 2009-2017 |
| KAN_U | 67.0003 | 47.0253 | 1840 | 2010-2017 |
| TAS_L | 65.6402 | 38.8987 | 250 | 2008-2017 |
| TAS_U | 65.6978 | 38.8668 | 570 | 2009-2017 |
| TAS_A | 65.779 | 38.8995 | 890 | 2014-2017 |
| MIT | 65.6922 | 37.828 | 440 | 2010-2017 |
| QAS_L | 61.0308 | 46.8493 | 280 | 2008-2017 |
| QAS_M | 61.0998 | 46.833 | 630 | 2017 |
| QAS_U | 61.1753 | 46.8195 | 900 | 2009-2017 |
| QAS_A | 61.243 | 46.7328 | 1000 | 2013-2014 |
| NUK_L | 64.4822 | 49.5358 | 530 | 2008-2017 |
| NUK_U | 64.5108 | 49.2692 | 1120 | 2008-2017 |
| NUK_K | 64.1623 | 51.3587 | 710 | 2015-2017 |
| NUK_N | 64.9452 | 49.885 | 920 | 2011-2014 |

**Table 2.** Metadata for the PROMICE automatic weather station network used in this work. Table adapted from Fausto et al. (2021) where the latitude, longitude, and elevation are derived from automated GPS measurements in summer 2016 or during the last weeks of operation if discontinued.

The best $p$ chromosomes are then kept and ranked, based on their cost function values. More weight is then given to the best parents for the next random selection. Further description of this algorithm applied to ORCHIDEE can be found in Bastrikov et al. (2018)'s comparative study.

### 2.3.2 Sensitivity analysis

With ORCHIDAS, it is also possible to perform a sensitivity analysis (SA) of the model. An SA tests the sensitivity of a model output (usually a physical variable). It tests how the output changes, with respect to different inputs - here the model

parameters. This is usually done before optimisation to ensure the right parameters and ranges of variation are used in the main experiments. In this study we use the Morris method (Morris, 1991; Campolongo et al., 2007), which is effective with relatively few model runs compared to other methods (e.g., Sobol', Sobol (2001)). Using an ensemble of parameter values, the Morris method determines incremental ratios, known as 'elementary effects', based on changing parameters one at a time in a sequence for many trajectories which populate parameter space. The mean ($\mu$) and standard deviation ($\sigma$) of the differences in model outputs for all the trajectories are calculated. This global method determines which parameters have a negligible impact on the model and which have linear and non-linear effects. The results of this method are qualitative, ranking the parameters in order of significance. To assess the results, we look at the normalised means, dividing through by $\mu$ of the most sensitive parameter. As such, the values we consider are between 0 and 1, with 1 representing the most sensitive parameters and 0 parameters with no sensitivity. Morris has also been previously used to test parameters for calibration of an earlier version of the ORCHIDEE snow model (Wang et al., 2013; Dantec-Nédélec et al., 2017).

### 2.3.3 Performance metrics

To assess the optimisation results, we rely on two standard metrics: the root-mean-square deviation (RMSD) and total absolute error (TAE),

$$\text{RMSD} = \sqrt{\frac{\sum_{i=1}^{n}[\mathbf{y}_i - M(\mathbf{x}_i)]^2}{n}}; \qquad \text{TAE} = \sum_{i=1}^{n}|\mathbf{y}_i - M(\mathbf{x}_i)| \tag{5}$$

where n is the total number of data points.

### 2.3.4 Posterior uncertainty

Assuming Gaussian prior errors and linearity of the model in the vicinity of the solution, the posterior error covariance matrix of the parameters, $\mathbf{A}$, can be approximated by

$$\mathbf{A} = \left[\mathbf{M}^T\mathbf{R}^{-1}\mathbf{M} + \mathbf{B}^{-1}\right]^{-1} \tag{6}$$

where $\mathbf{M}$ is the model sensitivity (Jacobian) at the minimum of $J(\mathbf{x})$ (Tarantola, 2005).

## 2.4 Experimental setup

### 2.4.1 Defining edges

The edges of the ice sheet are of particular interest since they correspond to areas of strong ablation and show the greatest variations in runoff and surface mass balance (SMB). To identify the edges of the GrIS, we exploited the fact that the edges are steeper than the middle of the ice sheet. To calculate the slope of a given pixel, we used the NOAA (National Oceanic and Atmospheric Administration) National Geophysical Data Center (NGDC) - ETOPO2 product (NOAA, 2006), which is based on a 2 arc-minute global relief model of Earth's surface and integrates land topography and ocean bathymetry. This product is

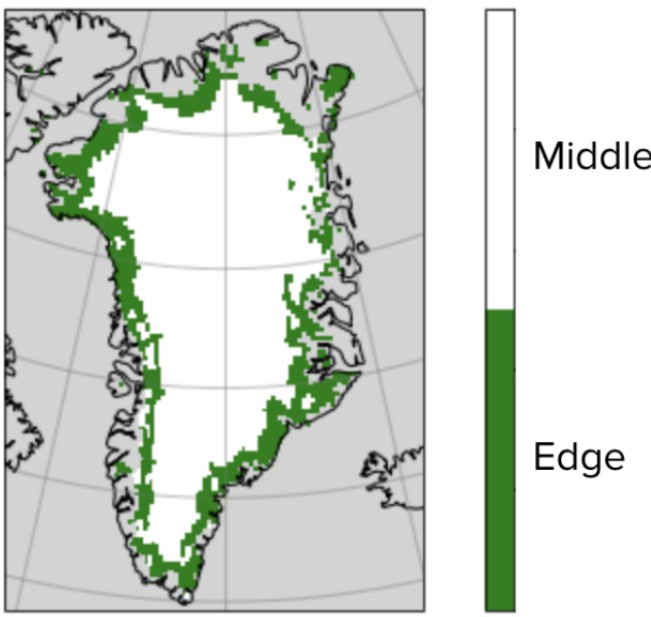

**Figure 1.** Spatial distribution of edge points (green) and middle points (white); selected based on the steepness of the pixel.

already integrated into ORCHIDEE, where it is used to determine the fraction of runoff that pools in flat areas (Ducharne, 2016; d'Orgeval et al., 2008). In a default ORCHIDEE simulation, when the slope is greater than 0.5%, all precipitation over that pixel that exceeds the infiltration capacity is run off immediately (Hortonian runoff); otherwise, it can pond at the soil surface and infiltrate at the next time step. Remember that each pixel in our Greenland simulations in this study has a resolution of 20 km and so the steepness of the slope applies over a large region. We found that by using this same threshold of 0.5%, we were able to encapsulate the edges of the GrIS (Fig. 1). As such, we refer to pixels with a slope gradient greater than 0.5% as "edge" points and the rest as "middle" points. These edge points account for just over 25% of all pixels. They were also the pixels with the largest errors when the standard ORCHIDEE run is compared to the retrieved MODIS snow albedo data; these edge pixels represented 78% of the pixels with RMSD greater than 0.1.

### 2.4.2 Experiments

ORCHIDEE was run over the whole GrIS with a spatial resolution of 20 km and a half-hourly time step, with a daily output frequency. The model was driven using meteorological data from MAR and confronted with MODIS albedo retrievals aggregated to the same resolution of 20 km. All the simulations performed in this study include two years of model spin-up to allow snow to accumulate. In each case, the two years preceding the years of study were used in the spinup and the model normally over these years (i.e., allowing for accumulation and melting) from an initial snow depth of 0. These two years are not included in calculating the cost function during the optimisations or during the analysis, but are important in ensuring correct initial

states. Furthermore, since during the winter months there is not enough solar illumination to compute the albedo, the months November to February are excluded from the optimisations and analyses.

For the main experiment, to capture the inter-annual variability of snow albedo, we selected three random years to perform our optimisation: 2000, 2010, and 2012. We optimised over these three years simultaneously. This means that, in this main experiment, we minimised a cost function comprising a sum of three cost functions, one for each year considered. The rest of the 2000-2017 time series was used for validation. During this main experiment, we optimised over the whole of the GrIS but gave an extra weight of four to the edge points (see Sect. 2.4.1). In early tests, we found that since the number of edge points

is being dwarfed by the much denser middle of the ice sheet, improvements were mainly concentrated over the middle of the ice sheet. This led us to choose to give extra weight to edge points during the main optimisation. The edge points account for approximately a quarter of the points. To ensure the edges and middle both contribute to the cost function, while also giving a bit more focus to the edge points, we chose to give an extra weight of four to the edges when calculating the cost function in the main optimisation. This main experiment, referred to as "Both", was complemented by two more optimisations: one just

over the edges of the ice sheet ("Edges") and one just over the middle points ("Middle"), again for the same three years. These were done to help analyse the posterior parameter values in Sect. 3.2.3. Finally, an additional experiment was performed to gauge the maximal improvement we could expect at the edges of the ice sheet. This was done to see whether the weighting used at the edges was sufficient, full details of which can be found in Appendix A. For each optimisation, 15 iterations of the genetic algorithm were used, which was enough for the system to converge.

To conclude the study, we performed a sensitivity analysis using Morris's method to understand the relative importance of the different model parameters in simulating albedo. In this experiment, we also considered additional parameters controlling the rate of density change and additional model outputs including SMB and runoff. These were included to better understand the relationship between different ice sheet processes and to identify which parameters and model output we might consider in future optimisations. This analysis compared ORCHIDEE outputs to the MAR model outputs, testing how each parameter

affected the RMSD between both models.

## 3    Results

### 3.1    Prior model

Before using ORCHIDAS to optimise the model parameters, the ORCHIDEE model was first tuned manually through trial and error. While not as robust as using a minimisation algorithm, this initial step is common for land surface modellers and helps

get a sense of the different parameter sensitivities. The primary focus of this manual tuning was to better capture the behaviour of the GrIS at its edges. This was achieved by increasing the overall albedo of fresh snow ($A_{aged} + B_{dec}$) and the snowfall depth required to reset the snow age ($\delta_c$), while also decreasing the albedo of aged snow and decreasing the rate of snow age decay ($\tau_{dec}$). Furthermore, one of the tuning constants for glaciated snow-covered areas was decreased ($\omega$). The rest of the parameters were kept as the default ORCHIDEE parameters (see Table B1 for full results).

This initial tuning helped the model to better simulate the albedo at the edges of the ice sheet, especially in the western part (Fig. 2), as well as other snow states such as SMB and runoff, which were also used to assess the success of the manual tuning. The tuned model was able to capture slightly more of the spatial variability of albedo in the middle of the ice sheet. Figure 2 also shows the albedo from the MAR product, the MAR product is used to drive ORCHIDEE and later to evaluate model performance. We can see that the MAR fits MODIS albedo better than the standard ORCHIDEE model. The overall RMSD

value for MAR is lower and the snow albedo is higher in magnitude, more closely matching MODIS. However, MAR shows less spatial variability - the albedo on the ice sheet looks uniform. The tuned version of ORCHIDEE does better than MAR, both in RMSD and spatial patterns. However, the north-south albedo gradient observed in the satellite retrievals was still not simulated, and overall, the albedo remains underestimated over the ice sheet. This initially tuned model was used as the prior for the albedo optimisation.

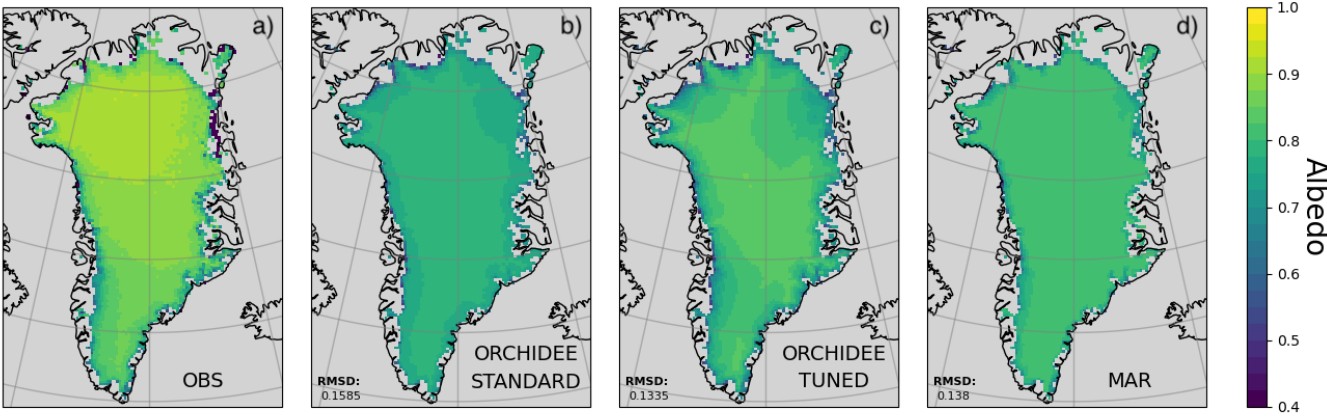

**Figure 2.** Retrieved and simulated mean albedo over Greenland (averaged over March-October for 2000-2017); a) shows the retrieved MODIS values, b) shows simulated albedo in the standard ORCHIDEE version (before tuning), c) shows the simulated albedo from the manually tuned model and d) shows albedo from the MAR model. The bottom left-hand corner of each panel shows the RMSD between modelled (ORCHIDEE or MAR) and observed (MODIS) albedo.

## 3.2 Main optimisation

### 3.2.1 Optimisation and validation

For the main optimisation, the GrIS albedo was optimised over the years 2000, 2010 and 2012 simultaneously, with a larger weight given to the edges (see Sect. 2.4.2 for the full setup description). Although a subset of three years was used in this optimisation, the improvement observed is consistent over all years (Figure 4a and Table 3). Indeed, some of the years with

270 the greatest reductions in RMSD were years not used in the optimisation e.g. 2003, 2009, and 2016. The troughs during the summer months are where the improvement is the most marked. The albedo during the summer months in prior simulations decreased too much. In the posterior run, these troughs more closely match the retrieved values.

a)

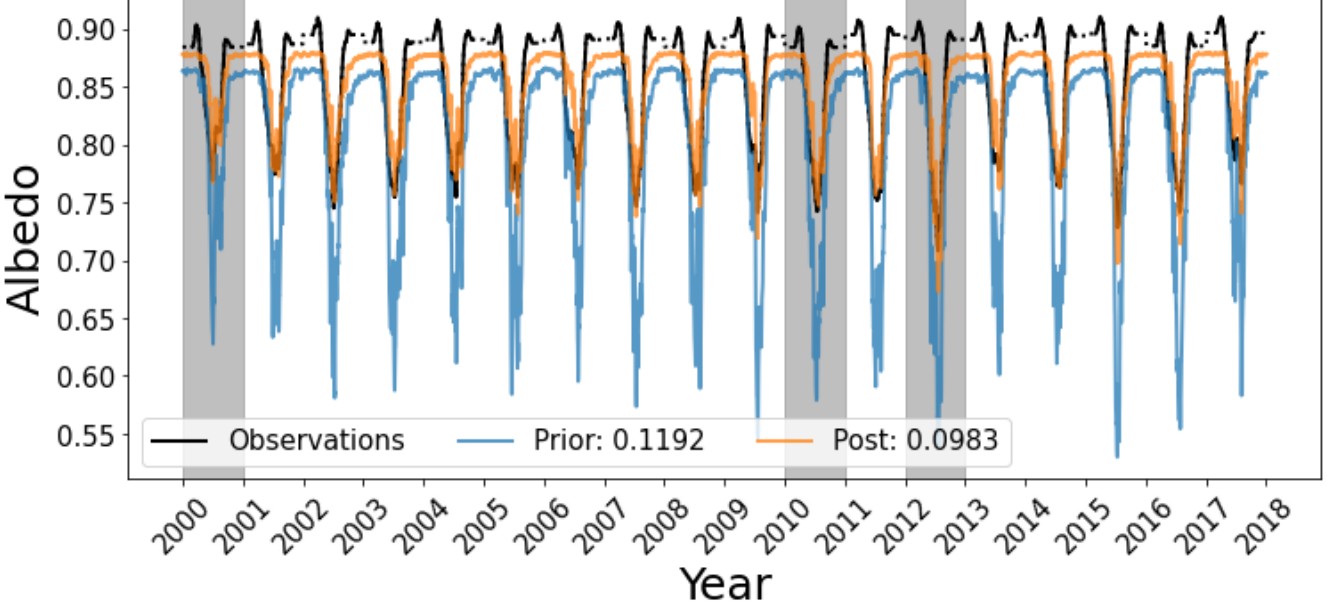

b)

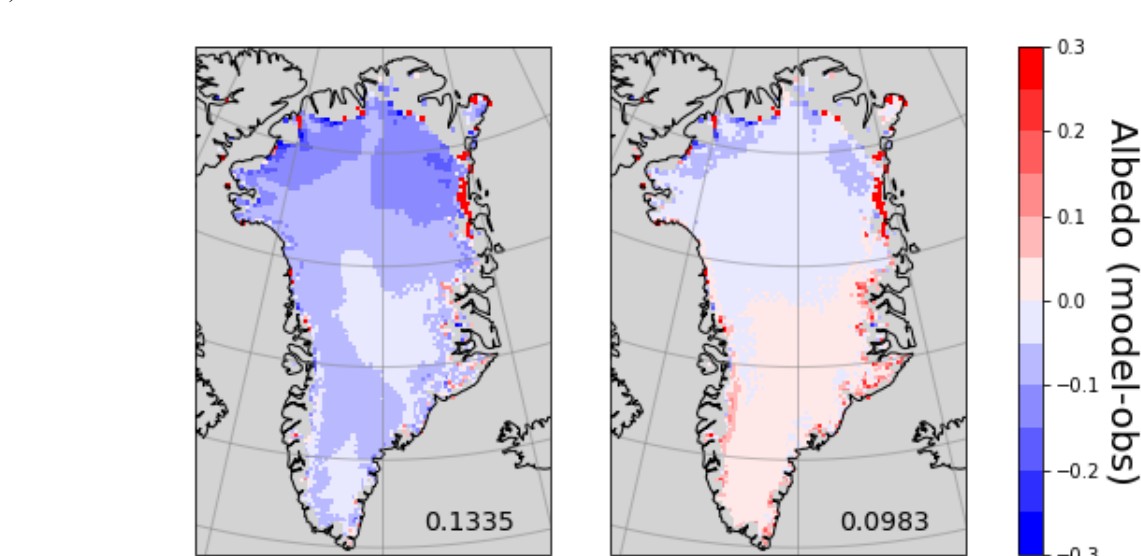

**Figure 3.** a) Time series of the snow albedo (averaged over space). The retrieved values (black), prior simulation (blue), and posterior simulation (orange), i.e. using the optimal parameter set (orange), are shown. The values in the legend denote the RMSD between each simulation and the retrieved albedo. b) Spatial distribution of differences between the model and the retrieved albedo averaged over March-October for the years 2000-2017 for both the prior (left) and posterior (right) models, with the total RMSD in the bottom right-hand corner.

When considering the errors of the posterior model spatially (Figure 4b), we noticed a slight underestimation of modelled albedo in the north of the ice sheet and a slight overestimation in the south. We also see that the edges are mostly overestimated. However, the RMSD reductions over the edge points are similar in magnitude to the reductions found in the preliminary optimisation where only the edge points were considered (Tables A1 and 3). This means that the weighting used between the edge and middle points during the optimisation was sufficient - we have achieved as low RMSD at the edges as in the edge-only experiment. By including the middle points in our optimisation, we greatly improve the fit of the model in the middle of the ice sheet - much more so than when only focusing on the edges (43.7% reduction compared to 8.51%). Figure 4 further illustrates where the error is reduced. By decomposing the TAE, we can see that both the edge and the middle points contribute to the error reduction. Figure 4 also allows us to compare the improvements between the different ORCHIDEE simulations. Note that the tuned model was used as the prior for the optimisation. The optimised model has the lowest error overall, both for the middle and the edges of the ice sheet. Figure 4 highlights the power of the ORCHIDAS approach - the total absolute error is reduced more substantially using the framework than when the manual tuning approach was used.

|      | Whole area | Edges | Middle |
| --- | --- | --- | --- |
| **2000** | **22.3** | **11.27** | **37.62** |
| 2001 | 25.73 | 11.22 | 43.36 |
| 2002 | 26.17 | 12.07 | 42.13 |
| 2003 | 28.89 | 12.39 | 44.65 |
| 2004 | 26.85 | 11.77 | 43.79 |
| 2005 | 27.08 | 9.38 | 45.36 |
| 2006 | 21.39 | 8.21 | 37.92 |
| 2007 | 26.55 | 6.49 | 46.06 |
| 2008 | 27.1 | 10.44 | 43.98 |
| 2009 | 29.17 | 11.75 | 45.61 |
| **2010** | **27.21** | **8.41** | **46.15** |
| 2011 | 27.31 | 6.65 | 46.46 |
| **2012** | **25.76** | **7.02** | **42.3** |
| 2013 | 25.0 | 6.54 | 43.61 |
| 2014 | 24.58 | 6.79 | 42.46 |
| 2015 | 27.35 | 10.19 | 43.09 |
| 2016 | 28.46 | 8.79 | 45.31 |
| 2017 | 26.04 | 11.7 | 41.9 |
| ALL | 26.37 | 9.52 | 43.68 |

**Table 3.** Percentage reduction in model-data RMSD between the prior and posterior runs over March-October. The years used in the optimisation are shown in bold.

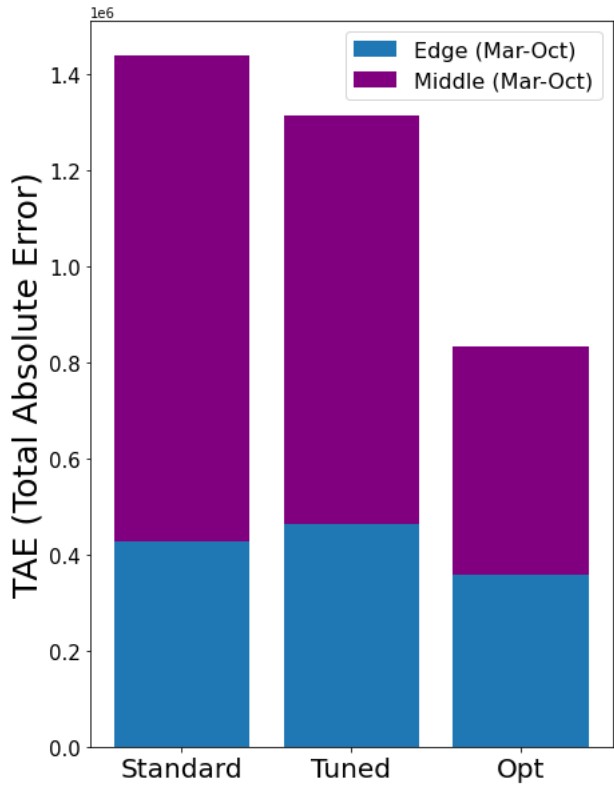

**Figure 4.** Total Absolute Error between the modelled and the retrieved MODIS albedo for the standard ORCHIDEE (i.e., default parameters values, left), the manually tuned (middle), and the optimised (i.e., using Bayesian framework, right) models. The Total Absolute Error is decomposed in each case, illustrating the contribution of the edge and middle points to the error for March-October.

### 3.2.2 Evaluation over PROMICE in situ sites

To evaluate the success of the optimisation, it is important to confront the results with data from a different source. Here we look at how the fit against albedo at in situ sites is improved with the optimisation (Fig. 5). Generally, the albedo is found to improve. The fit to the observations results in a lower RMSD compared to when using the prior model. With the exception of UPE, reductions in RMSD are greater for the upper sites (between 11 and 25%) than for the lower sites (between -6 and 8%, where negative means the fit has degraded). For the UPE sites, this is the opposite. Of the 24 sites tested, the fit to the observations is only degraded in three cases. These sites are all lower sites - i.e., where the measurement station is near the ice sheet margin, where processes are harder to model. Two sites are found on the eastern edge of the ice sheet (SCO_L, TAS_L), and the last one is found at the southern tip of the ice sheet (QAS_L). When comparing to Fig. 3b, we can see that the eastern

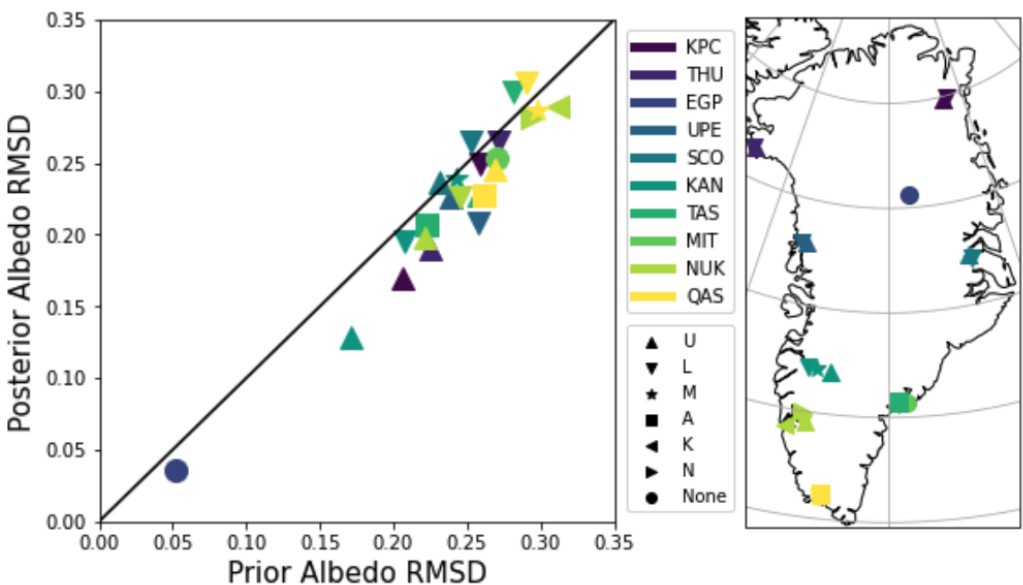

**Figure 5.** Evaluation of model-observation fit over PROMICE sites. For each year of available data, the RMSD for the months (Mar-Oct) is calculated. Different colours represent different sets of sites and the shapes represent the subscript used to identify individual sites (see Table 2). The mean over these RMSD values is shown in the figure. Points below the 1-to-1 line represent sites where the model-data fit is improved by the optimisation.

edge of the ice sheet is where the largest errors occur, even after the optimisation. Furthermore, TAS_L and QAS_L are two locations where the smallest amplitude and highest winter temperatures occur (van As et al., 2011, Fig.1) due to being exposed to the relatively warm wintertime atmospheric conditions of the Atlantic Ocean.

    Figure 5 also shows us how ORCHIDEE generally performs at these sites - the magnitude of the RMSD remains similar for both parameter sets. Since the sites are mainly found at the edges of the ice sheet, errors are generally high - between 0.15 and

0.32. The two sites with the lowest RMSD for both the prior and posterior models are the ones located near the middle of the ice sheet, in the accumulation area (KAN_U and EGP). There is no obvious link between latitude and the magnitude of the errors. Instead, elevation due to the position on the edges of the ice sheet is a more important factor.

    Overall, this evaluation is encouraging - it shows that the optimisation was successful at improving model albedo when tested against a different data source. Nevertheless, we do need to highlight a couple of shortcomings in this comparison. Firstly, we

do not have accurate local forcing data at the sites with which to drive ORCHIDEE. Therefore, the 20km MAR data was used, meaning that we are comparing observations and the model at different resolutions. Secondly, MODIS has been validated, and some of its biases due to the solar zenith angle were corrected for, using PROMICE data (see Sect. 2.2.2). As such, the MODIS data used in the optimisation is not completely independent from the PROMICE data used in this evaluation.

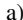

a)

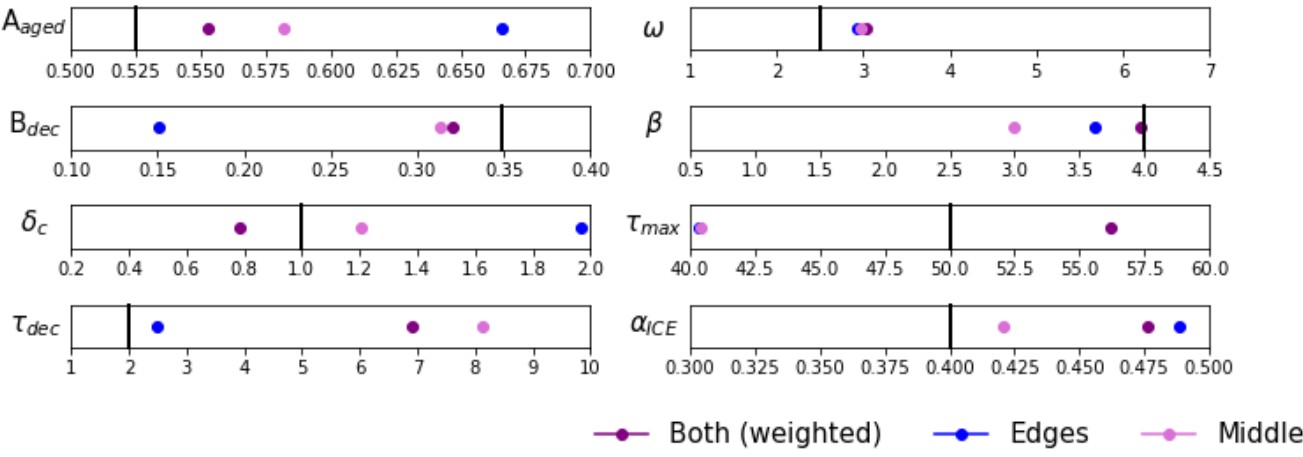

Both (weighted) ——•—— Edges ——•—— Middle

b)

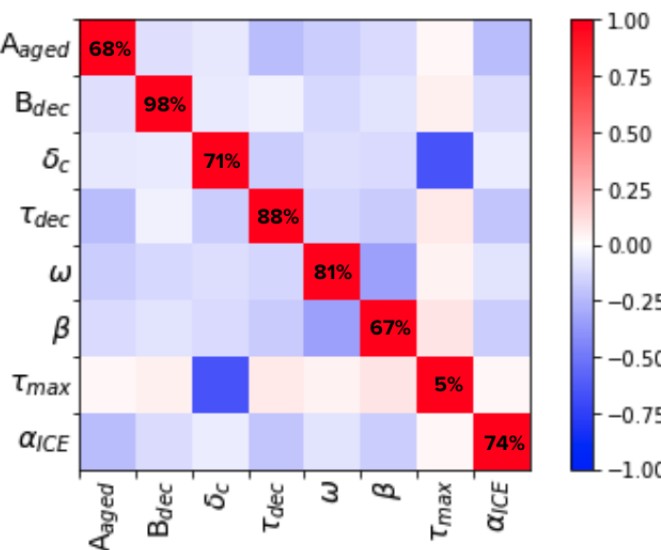

**Figure 6.** a) Posterior parameter values found for three different optimisations; "Both" where the middle and edge points are weighted with a ratio of 1:4, "Edges" where only the edge points were used in the optimisation, and "Middles" where only the middle points were used. Each box's range represents the variation used for each parameter during the optimisation. The vertical black line represents the prior parameter value. b) Correlations between the posterior parameters calculated at the optimum of the "Both" optimisation. Percentages on the diagonal indicated in the reduction in parameter uncertainty also calculated at the optimum (see Sect. 2.3.4).

### 3.2.3 Posterior parameters

In this section, we consider how the parameter values have changed to fix the model-data disparities. In Fig. 6a, we look at the posterior parameters from the main experiment (referred to as "Both") and posterior parameters from experiments solely optimising the edge points ("Edges") and solely optimising the middle points ("Middle"). Initially, the prior model underestimated the albedo. This underestimation is seen both temporally (Fig. 6a), where the maximum simulated albedo is below that of the retrieved values, and spatially (Fig. 2), where the underestimation is most noticeable over the centre of the ice

sheet. For all three optimisations, $A_{aged}$ and $\alpha_{ICE}$ increase, contributing to fixing this underestimation. These two parameters directly impact the albedo - as they increase, so will the albedo of the GrIS. We also saw that in the prior model, the albedo decayed too much in summer (Fig. 3a). In the posterior models, the value of the $B_{dec}$ parameter is lowered, giving less weight to the decay term in Eq. 1. Again, this decrease occurs for all three optimisations. Similarly, $\tau_{dec}$ increases in all cases, which also leads to a smaller decay term. Finally, we see that $\omega$ values increase and $\beta$ values decrease. By doing so, these two parameters

increase the value of $g_{temp}$ which appears in the denominator of $f_{age}$ (Eq. 3) hence slowing down snow ageing.

We also notice some differences between the three sets of posterior parameters. Since the "Both" optimisation includes points from both of the other optimisations, we might expect the posterior parameters to be in between the "Edges" and "Middle" posterior parameter values acting as a compromise between both optimisations. However, this is only true for two out of the eight parameters. Instead, the "Both" posterior parameters often take higher or lower values than parameters from

325 the other two optimisations. This behaviour suggests that parameter space is not smooth but full of local minima. The clearest example of the "Both" optimisation performing differently is for the parameters $\delta_c$ and $\tau_{max}$. These increase and decrease respectively for the "Edges" and "Middle" optimisations. However, for the "Both" optimisation, the opposite is true. These parameters can be highly anti-correlated (Fig. 6b). If $\delta_c$ is very small, the snow's age does not reset to zero, so the snow ages for longer, necessitating a larger value of $\tau_{max}$. Therefore, these two parameters, $\delta_c$ and $\tau_{max}$, compensate for each other.

However, this relationship is seen to not be critical when we consider the variance at the optimum. We can see that $\tau_{max}$ remains unconstrained by the optimisation. The reduction parameter uncertainty is small - the lowest of all the parameters. The other parameters show high levels of parameter uncertainty reduction, showing they are highly contained by the optimisation, with $B_{dec}$ reducing the most.

### 3.3 Impact of the different parameter sets on modelling the surface mass balance of the Greenland Ice Sheet

### 3.3.1 Comparison between ORCHIDEE and MAR model outputs

In Fig. 7 and 8, we consider how the different parameter sets discussed in this study impact the modelled snow states. To assess the performance of the different ORCHIDEE parameter sets, we compare the model outputs to that of the MAR model. Although MAR is a model with its own biases and errors, it has been shown to have good estimations of the different snow states (Fettweis et al., 2017, 2020) and so is a good product against which to compare.

In particular, we are interested in better modelling the surface mass balance (SMB) and its components (sublimation and runoff). SMB measures the difference between mass gains and ablation processes, hence dominating the rates of mass change

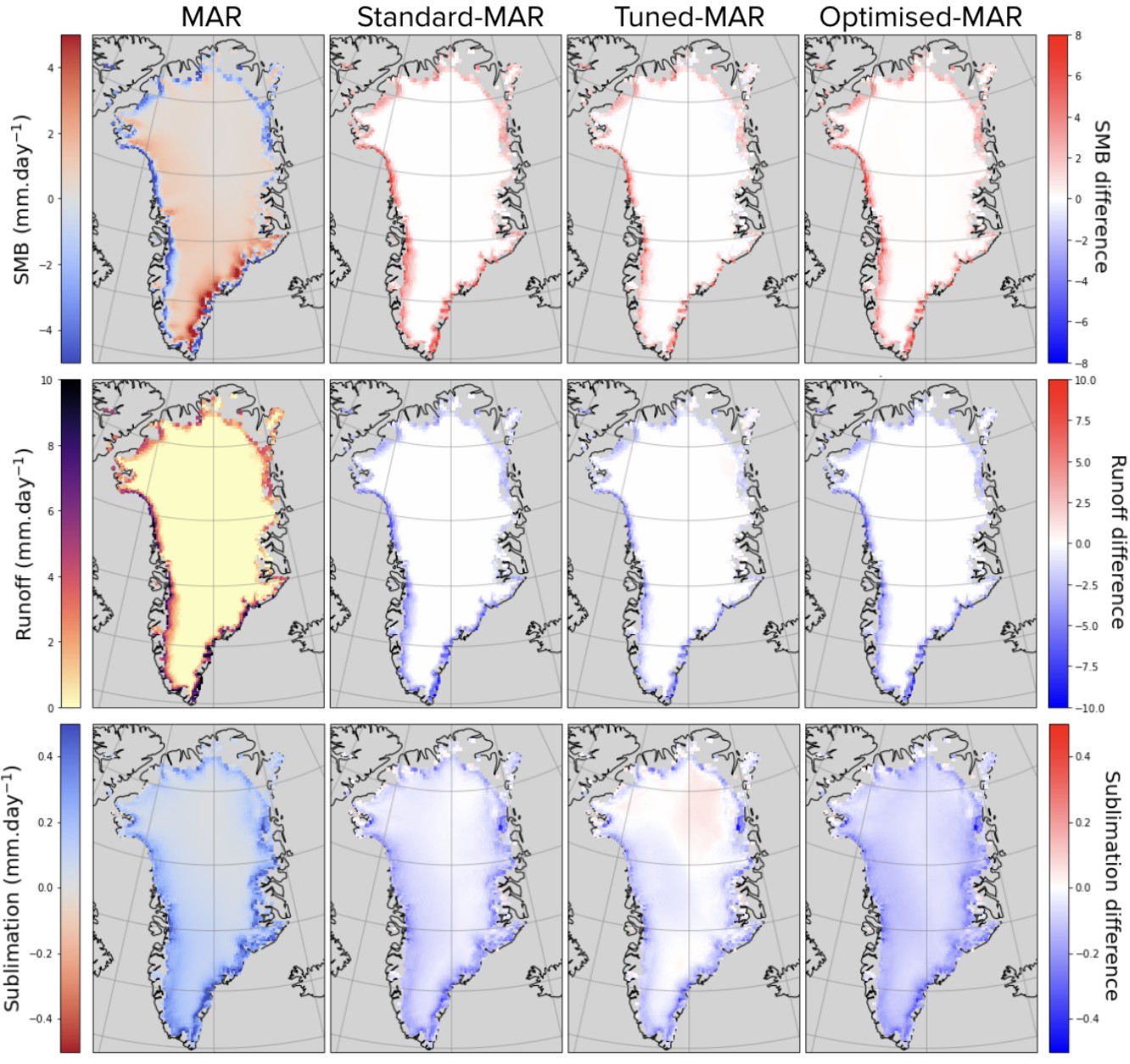

**Figure 7.** Impact of different parameter sets on ORCHIDEE simulations; "Standard" uses default parameter values, "Tuned" uses parameter values from the manual tuning and "Optimised" from the ORCHIDAS optimisation. Shown are spatial maps averaged over time (March-October) for MAR (left) and the difference between ORCHIDEE and MAR. Each row features a different variable of interest (Top: SMB, Middle: Runoff, Bottom: Sublimation).

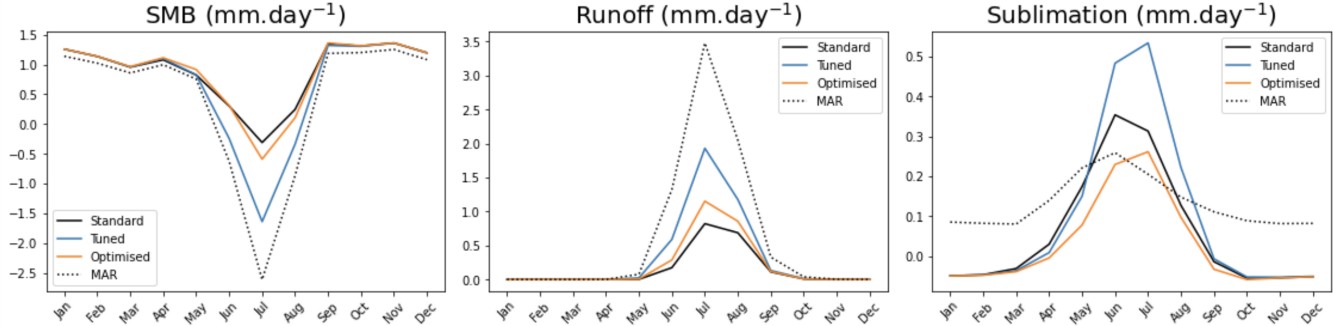

**Figure 8.** Same as Fig. 7 but showing monthly means averaged over space. This time the columns feature the different variables of interest.

over the GrIS. The manually tuned version of ORCHIDEE simulates SMB the closest to MAR's SMB. This can be seen both spatially and temporally. Spatially, the differences between MAR and the ORCHIDEE simulations are observed at the edges - especially in the north and west of the GrIS. The most noticeable difference in the ORCHIDEE runs can be seen at the
345 west of the ice sheet, where the tuned model simulates SMB the best when compared to MAR, followed by the optimised model. In both the manually tuned and optimised runs, the SMB is reduced at the west of the ice sheet compared to the default ORCHIDEE run. This is mirrored by an increase in runoff at the western edge of the ice sheet. Indeed, for simulated runoff, changes are mainly found at the western edge of the ice sheet, with the tuned model performing the best and the optimised model second best when compared to MAR. Both parameter sets (optimised and tuned) improve the fit compared to the default
ORCHIDEE simulations. However, neither is able to capture the magnitude of the runoff in summer, with the tuned model still only simulating half the expected magnitude of runoff.

When we consider modelled sublimation, the differences between each ORCHIDEE simulation are most marked. By increasing the albedo over the ice sheet, we decrease latent heat over the area and hence sublimation. When considering the time series, we see that the optimised model gets the correct magnitude of sublimation during the summer months. All of the
355 ORCHIDEE simulations have a delayed peak compared to MAR and no sublimation is simulated by ORCHIDEE outside the summer months. Indeed in winter, we even get negative values, i.e., condensation. This is likely due to the fact that surface temperatures are generally lower than those from MAR, leading to a lower saturation humidity and, thus, to condensation. When averaged over time, we see that MAR has high sublimation rates at the eastern edge of the GrIS. However, none of the ORCHIDEE simulations capture this. Instead, the sublimation over the centre of the ice sheet is what changes with the
360 different parameter sets - with the optimised model lowering the rates the most. The strong impact that changing albedo has on simulated sublimation over the whole of the GrIS shows how coupled albedo and sublimation are in the model.

Overall, with the optimised model, we do better than the standard ORCHIDEE model but not as well as the tuned model. During the manual tuning of the albedo parameters, the performance of the new parameters was assessed against several model outputs, including SMB, sublimation and runoff at each step of the trial and error procedure. We can think of this manual tuning
as a multi-objective calibration. When performing the optimisation, we get the best fit to the albedo. However, we overfitted to

albedo with no other data, degrading the fit to other model outputs. As seen with the posterior parameters, parameter space is not smooth but has many local minima. As such, it is possible that a different solution exists, reducing the albedo to a similar extent whilst also improving the fit to other modelled outputs. To achieve this, we need to include more data in the optimisation to perform a multi-objective optimisation. If we cannot find such a parameter set, this would point to structural problems in the model, i.e., missing processes. The fact that MAR has a more complex snow model that works better at capturing the different processes over Greenland leads us to believe structural changes are needed in ORCHIDEE for it to be able to better simulate SMB and its components. Through the optimisation, we have improved the representation of albedo but not of SMB and its components. This is because albedo is not the only important parameter in the modelling of the snowpack evolution. Other processes like melting depend on the snow's temperature profile, compaction, and refreezing, therefore on the thermal and mechanical properties of the snowpack. These processes must be well represented in the model and may require further calibration in future works.

### 3.3.2   Sensitivity analysis of ORCHIDEE parameters

In any parameter estimation study, performing a preliminary sensitivity analysis to select the parameters for the optimisation is standard practice. Since the albedo parameterisation had a manageable number of parameters, we proceeded directly to the optimisation. However, since the different processes of the snow model are interlinked, we decided to perform a sensitivity analysis to conclude this study. In addition to understanding the different sensitivities, this was done to help understand how other simulated quantities are also affected by the albedo parameters, notably SMB and its components, and to highlight which further parameterisations to consider in future experiments. This is especially important if we were to optimise the snow model against other types of observations either individually or simultaneously with the albedo retrievals. We add parameters from two other parameterisations controlling snow viscosity and settling freshly fallen snow (described in Sect. B2) to better understand the relative importance of the different parameters.

   Parameters from the albedo parameterisation significantly affect all simulated outputs tested in this sensitivity analysis. For the simulated albedo, the most sensitive parameter is $B_{dec}$ for both the middle and edge of the ice sheet (Fig. 9). This is consistent with the reduction in parameter uncertainty found in Fig. 6b, which was the highest of all the parameters optimised. We also see that the heat fluxes, surface temperature, and sublimation in the middle of the ice sheet are sensitive to $B_{dec}$. In addition, the parameter controlling the snow decay rate ($\tau_{dec}$) is the most sensitive parameter for simulating sublimation and the latent heat flux over the whole ice sheet (Fig. 9), and one of the most sensitive for sensible heat flux. Since both $B_{dec}$ and $\tau_{dec}$ control the impact of snow decay, they directly impact the albedo of the snow and, therefore, the surface temperature. The surface temperature directly affects runoff and the sensible heat flux (calculated as a function of the difference between the surface temperature and the temperature of the atmosphere). The latent heat flux depends directly on the snow, ice and bare soil fractions. The higher the amount of runoff, the more likely it is to have areas where all the snow melts (or grid points where the snow fraction decreases). Therefore the latent heat flux on the snow decreases and so does the sublimation.

   The model outputs are only marginally sensitive to $\tau_{max}$. Since we normalise the Morris score by the highest ranking parameter, this shows that compared to the most sensitive parameter, $\tau_{max}$ is the least important albedo model parameter in

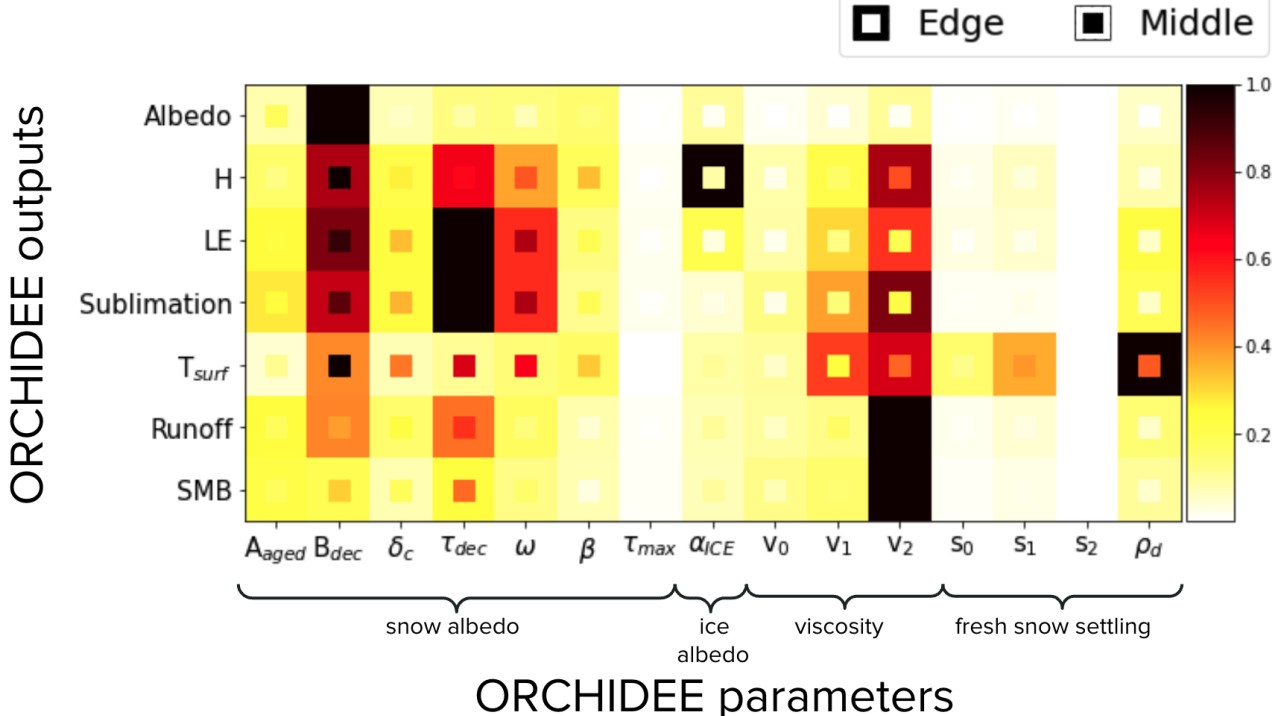

**Figure 9.** Heatmap showing the relative sensitivity of each parameter for different simulated model outputs; albedo, sensible heat flux (H), latent heat flux (LE), sublimation, surface temperature ($T_{\text{surf}}$), runoff, and surface mass balance (SMB). In each case, the sensitivity of the parameters is shown for simulated quantities at the edge of the ice sheet (shown by the filling at the edge of each box) and in the middle of the ice sheet (shown by the filling in the middle of each box). Morris scores (see Sect. 2.3.2 for discussion of Morris scores) are normalised by the highest ranking parameter in each case. Dark squares represent the most sensitive parameters for each output, and light squares represent parameters with little to no sensitivity.

explaining the possible range of responses for each modelled output tested. This is again consistent with the optimisation results in Sect. 3.2.3, which found $\tau_{max}$ to be the least constrained by the optimisation. Although seen to be correlated to $\delta_c$ at the optimum of the cost function (Fig. 6b), changes in $\delta_c$ have more impact on the model outputs than $\tau_{max}$, especially at the centre of the ice sheet. Since $\delta_c$ appears in the exponential term of Eq. 3, small variations in its value will have a larger impact on the snow age $\tau_{snow}$ than small variations in $\tau_{max}$. Furthermore, high uncertainty remaining around the $\tau_{max}$ parameter at the optimum implies that this relationship is not critical in the snow model.

The last two parameters of the albedo parameterisation, $\omega$ and $\beta$, can be seen to impact temperature and the sensible heat flux at the centre of the ice sheet. These parameters are present in the part of the parameterisation controlling the effect of low temperature on metamorphism (Eq. 3). By influencing snow ageing, these parameters impact surface temperature (through changes in albedo) and thus the sensible heat flux.

The sensible heat flux is especially sensitive to the parameter determining the ice albedo ($\alpha_{ICE}$) at the edges of the ice sheet. We expect the snow to melt faster at the edges exposing the bare ice below and hence increasing the importance of ice albedo. The ice albedo will therefore impact the surface temperature at these exposed edge points and thus the sensible heat flux.

Modelled albedo is not very sensitive to parameters from the viscosity and fresh snow settling parameterisations - especially not at the centre of the ice sheet. However, these parameters are important for other modelled quantities. The runoff, surface mass balance, and sublimation are sensitive to the viscosity parameters (Eq. B2). The parameter controlling the impact of snow density in this parameterisation ($v_2$) is the most sensitive. When viscosity decreases, snow density increases and liquid water holding capacity decreases. This leads to an increase in runoff and a decrease in SMB. If the increase in runoff at the edges leads to a significant decrease in snow cover, this will also impact sublimation (which depends on the snow fraction and temperature).

The ice sheet temperature at the surface is sensitive to fresh snow settling parameters (Eq. B3), especially to $\rho_d$, which is a parameter impacting snow density ($\rho_{snow}$). When considering the rate of density change equation (Eq. B1), we can see it comprises two terms: a term representing the compaction due to snow load and a term parameterising the effect of metamorphism, which is significant for fresh settling snow. With newly fallen snow, $\rho_{snow}$ is generally low (50-200 kg.m$^{-1}$), especially in cold environments with little wind and, therefore, very little drifting of snow. Depending on the value of $\rho_d$, the density term in Eq. B3 will become zero more or less quickly, maximising the value of $\psi_{snow}$. This, in turn, increases the density of snow ($\rho_{snow}$) in the model. As the density of snow increases, the snow becomes less insulating, and the thermal conductivity inside the snowpack increases. In other words, the temperature inside and at the snowpack's surface depends directly on the snow density. This sensitivity to the fresh snow settling parameters may be more important at the edges of the ice sheet because there is more precipitation than in the centre, where the climate is colder and, therefore, drier.

When comparing different ORCHIDEE runs to MAR (Sect. 3.3.1), we saw that sublimation was the output most impacted by the different parameter sets. This was especially notable at the centre of the ice sheet. This sensitivity analysis highlights that sublimation at the centre of the ice sheet is most sensitive to the B$_{dec}$ and $\tau_{dec}$ parameters, which are changed in the optimisation to lower the decay term and therefore increase albedo. In contrast, for runoff and SMB, both of which show no spatial variability over the middle of the ice sheet in Fig. 7, the $v_2$ parameter from the viscosity parameterisation is more important. However, this parameter was not optimised in this study. Nor were other parameters from the viscosity parameterizations, to which sublimation, runoff and SMB are sensitive, especially at the edges. Although we do get some variation in runoff and SMB in the different ORCHIDEE runs (Sect. 3.3.1), since these are concentrated at the edges, it is possible that by optimising these viscosity parameters we would better fit MAR outputs.

Overall, although modelled albedo is not very sensitive to parameters from the snow viscosity and settling of freshly fallen snow functions, parameters from these latter two parameterisations greatly impact the other model outputs tested. Especially the parameters from the viscosity parameterisation. Therefore, for future experiments, this sensitivity analysis suggests that to optimise energy budget, runoff and sublimation simultaneously, we would need to consider including the parameters from the albedo and viscosity parameterisations.

## 4 Discussion and conclusions

We have shown that by giving extra weight to the edge points during the optimisation, we can find a set of parameters that improves model-data fit for all the GrIS. The reduction of RMSD at the edges was similar to the reduction found when only focusing on the edge points during the optimisation. However, by including the middle points in the optimisation, the whole ice sheet greatly improved its fit to retrieved albedo. The model was optimised against three separate years simultaneously and validated against the rest of the time series. Improvements were consistent over all the years considered. We also evaluated the optimisation using in situ albedo with the PROMICE network with promising results - the RMSD at 21 out of 24 sites improved compared to the prior model. Further work will include testing the application of this model and parameters to other polar and non-polar regions, starting with other ice sheets such as Antarctica.

Parameter optimisation is a valuable tool for model development. Not only can it be used to find the best set of parameters for a given parameterisation, but more importantly, it can help identify structural issues in the model. When we cannot further improve the model against the observations, this can point to structural deficiencies in the model. For example, we cannot capture the different albedos in the north and south of the ice sheet with the current processes represented. More structural changes may help capture this variability. For example, we could look at further improving the snow/ice transfer processes by better discretising the snowpack vertically(Charbit et al., in prep.). Processes linked to the darkening on the ice sheet (e.g., deposition of aerosols, algae and dust) also need to be considered in future developments of the model. Since we are running the ORCHIDEE offline - i.e., prescribing the meteorological forcing - it would also be beneficial to run the model with different forcings to separate model structural errors from the errors in the forcing. This is important since MAR is a modelled estimate and, therefore, will be subject to its own biases and errors. We would want to ensure that we are correcting errors in the land surface model and not correcting atmospheric biases in the forcing data.

We must also remember that there are errors linked to the retrievals of the albedo from the observed quantity. Indeed, the large uncertainties in the winter months led us to omit them for this study. For the other months, we set the observation errors to be the mean-squared difference between the observations and the prior model simulation to also account for the structural model errors. However, in practice, the true errors may be very different. For example, although steps to correct the solar zenith angle bias in the product have been undertaken, it is possible that the strength of the north-south albedo gradient observed in the data is an artefact of the product. Without clear and robust uncertainty quantification, we cannot disentangle natural GrIS processes from biases in the retrievals. There is an urgent need for data producers to provide this uncertainty, ideally at each time step (Merchant et al., 2017).

In our optimisations, we put great importance on the edge points. However, these are also the points where we are most likely to find bare soil and vegetation instead of ice. These points could be represented by some of the other plant functional types in the model, which have different parameter values for $A_{aged}$ and $B_{dec}$. To identify and separate these pixels from the ice-covered pixels used in this study, future experiments could exploit the ESA CCI (European Space Agency Climate Change Initiative) land cover product (ESA, 2017), allowing us to optimise these parameters for each of the plant functional types present. For the optimisations, we also selected three random years instead of the full time series. However, it is possible that a different

subset of years would give different results. Nevertheless, given the consistent improvement found over the whole period, we do not think that the results would be too different.

We have also shown that while significantly improving the model's fit to retrieved albedo measurements, changing the parameters also influences the other model outputs. This was first done by considering the influence of the optimised parameters on other model outputs by comparing simulated snow states to the MAR model. The optimised model was found to perform more consistently with MAR outputs than the original ORCHIDEE model but not as well as the tuned model for simulating SMB and runoff. For sublimation, the optimised model simulated the most accurate magnitude in summer; however, it still 485 showed a bias when considered spatially. We also performed a Morris sensitivity analysis using a wider set of parameters. Morris was chosen since it only required a small number of model runs. However, its main limitation is that the sensitivity measure is only qualitative - the parameters are only ranked in order of significance but we do not quantify their absolute contribution. Furthermore, with this method, it is not possible to distinguish the nonlinear effect individual parameters have on the model output from the effect of their interactions with other parameters. It is also very dependent on the range of variations 490 assigned to the parameters. Nevertheless, the Morris approach can still help give a broad overview of the most influential parameters and the model outputs they impact.

Therefore, in addition to considering further structural changes, it will be necessary to further optimise the model's internal parameters against a range of datasets. With the ever-growing quantity of satellite datasets available, we could consider many different avenues. For example, we could use GRACE (Gravity Recovery and Climate Experiment) satellite mission data to 495 constrain SMB (Sasgen et al., 2020). To constrain ice velocity, we could use products based on Sentinel-1 retrievals (Mouginot et al., 2017; Andersen et al., 2020) and data from the ESA CCI land surface temperature project (Karagali et al., 2022) could be used to constrain surface temperatures. Combining these datasets with MODIS albedo would result in a rich data source to optimise the model's internal parameters and learn about different processes governing the ice sheet.

*Code availability.* The ORCHIDEE vAR6 model code and documentation are publicly available via the ORCHIDEE wiki page (http://
forge.ipsl.jussieu.fr/orchidee/browser/) under the CeCILL license (http://www.cecill.info/index.en.html, CeCILL, 2020). This is the version used in CMIP6. The associated ORCHIEE documentation can be found at https://forge.ipsl.jussieu.fr/orchidee/wiki/Documentation. The ORCHIDEE model code is written in Fortran90 and is maintained and developed under an SVN version control system at the Institute Pierre Simon Laplace (IPSL) in France. The ORCHIDAS data assimilation scheme (in Python) is available through a dedicated web site (https://orchidas.lsce.ipsl.fr).

**Appendix A:  Weighting the edge of the ice sheet**

To see what the maximal improvement in model-data fit we can expect over these edges, we performed a preliminary experiment optimising only these points for the months March-October (Table A1). We were able to reduce the RMSD at these edge points by approximately 10%. This optimisation was also able overall to improve the simulated albedo in the middle of the ice sheet in summer. This implies there is some consistency between the edge and middle points for the 2000 - 2017 period.

However, this optimisation did not improve the middle points consistently - for example, we observe a degradation in fit for the year 2000.

**Table A1.** Results of a preliminary experiment optimising only the edge points of the GrIS for March-October of 2000. The optimisation was performed using the GA algorithm. Percentage reduction of model-data RMSD. Negative numbers show an increase in RMSD i.e. a degradation in fit.

| Year | Edge points | Middle points | All points |
|------|-------------|---------------|------------|
| 2000 | 11.86 | -6.01 | 3.14 |
| 2000-2017 | 10.11 | 8.51 | 9.21 |

## Appendix B: Parameter information

### B1 Parameter values

In Table B1, we list the different parameter values used and found in this study.

**Table B1.** Parameters of the snow albedo model. Default values refer to parameters used in a standard ORCHIDEE simulation, tuned parameters refer to values found after the manual tuning experiments, and the optimised parameters refer to parameters values found after using ORCHIDAS.

| Parameter | Description | Default | Manually tuned | Optimised |
|-----------|-------------|---------|----------------|-----------|
| $A_{aged}$ | Sum to be the albedo of fresh snow | 0.62 | 0.525 | 0.553 |
| $B_{dec}$ |  | 0.169 | 0.349 | 0.320 |
| $\delta_c$ | Snowfall depth required to reset the snow age (m) | 0.2 | 1 | 0.783 |
| $\tau_{dec}$ | Snow age decay rate (days) | 10 | 2 | 6.911 |
| $\omega$ | Tuning constants for glaciated snow covered areas | 7 | 2.5 | 3.037 |
| $\beta$ |  | 4 | 4 | 3.974 |
| $\tau_{max}$ | Maximum snow age | 50 | 50 | 56.183 |
| $\alpha_{ICE}$ | Ice albedo | 0.4 | 0.4 | 0.476 |

### B2 Additional parameters

To get a better overview of the model output sensitivities, we consider addition parameters used to calculate the local rate of density change in the $i^{th}$ layer of the snowpack:

$$\frac{1}{\rho_{snow}(i)}\frac{\delta\rho_{snow}(i)}{\delta t} = \frac{g.\mathcal{M}(i)}{\eta(i)} + \psi(i) \tag{B1}$$

The first term, represents the compaction due to snow load. This depends on the pressure of the overlying snow, calculated
using the gravitational constant ($g$; m.s$^{-2}$) and the cumulative snow mass ($\mathcal{M}$; kg.m$^{-2}$) and snow viscosity ($\eta$). The second

term describes the effect of metamorphism ($\psi$), which can also be thought of as determining the settling of freshly fallen snow since this effect is most significant for newly fallen snow. Both the snow viscosity ($\eta$) and settling of freshly fallen snow ($\psi$) are solved in ORCHIDEE using the following empirical exponential functions of snow density ($\rho_{snow}$) and temperature ($T_{snow}$):

$$\eta(i) = \mathbf{v_0} \exp(\mathbf{v_1}(T_f - T_{snow}(i)) + \mathbf{v_2}\rho_{snow}(i)), \tag{B2}$$

$$\psi(i) = \mathbf{s_0} \exp(-\mathbf{a_1}(T_f - T_{snow}(i)) - \mathbf{s_2}(\max(0, \rho_{snow}(i) - \boldsymbol{\rho_d})). \tag{B3}$$

where $T_f$ is the triple-point temperature for water. The rest are parameters whose values and ranges of variation used in the sensitivity analysis are outlined in Table B2.

**Table B2.** Parameters used to calculate the local rate of density change. The default value refers to the value used in a standard ORCHIDEE simulation, min and max refer to the ranges over which the parameters are allowed to vart during out experiments.

| Equation | | Parameter | Units | Default | Min | Max |
|---|---|---|---|---|---|---|
| $\eta$ | (Eq. B2) | $v_0$ | Pa s | $3.7 \times 10^{-7}$ | $1.5 \times 10^{-7}$ | $4 \times 10^{-7}$ |
| | | $v_1$ | $K^{-1}$ | 0.081 | 0.08 | 0.35 |
| | | $v_2$ | $m^3.kg^{-1}$ | 0.018 | 0.009 | 0.02 |
| $\psi$ | (Eq. B3) | $s_0$ | $s^{-1}$ | $2.8 \times 10^{-6}$ | $1.5 \times 10^{-6}$ | $3.5 \times 10^{-6}$ |
| | | $s_1$ | $K^{-1}$ | 0.04 | 0.01 | 0.1 |
| | | $s_2$ | $m^3.kg^{-1}$ | 460 | 320 | 600 |
| | | $\rho_d$ | $km.m^{-3}$ | 150 | 100 | 200 |

*Author contributions.* SC and CD developed the snow model for its application over the GrIS, with support from FM and CO. VB developed the ORCHIDAS system and, with NR, expanded its application over 2D surfaces. NR integrated the sensitivity analyses to ORCHIDAS. Prior model tuning was performed by SC and CD. NR performed the optimisations and sensitivity experiments. NR generated the figures. All authors contributed to analysing the results, and writing the manuscript.

*Competing interests.* We declare that no competing interests are present.

*Acknowledgements.* Nina Raoult is funded by the European Space Agency (ESA) as part of the Climate Change Initiative (CCI) fellowship (ESA ESRIN/Contract No. 4000133601). We would like to thank the ORCHIDEE Project Team for developing and maintaining the ORCHIDEE code.

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
