# Peer review of "Improving modelled albedo over the Greenland ice sheet through parameter optimisation and MODIS snow albedo retrievals"

_EGUsphere, 2022_

## Referee Comment (RC1)

This paper deals with improving how surface albedo is represented in land surface models by data assimilation of satellite-derived albedo. I will note that I am not an expert on data assimilation and thus, cannot really comment on the applicability of the DA approach used here. The only comment I have about the DA is I'm not sure how all the parameters in Table 1 are tuned to fit the albedo, and maybe some more explanation as to how that is done is warranted. I find the sensitivity analysis ok so no strong comments there. My comments mostly pertain to the MODIS data and result of clarify in the manuscript.

One general comment, it is true that under low sun angles, the albedo retrieved from satellite is less accurate, yet I find it strange to have a discussion on winter months when the sun is below the horizon. I realize that there is a need to spin the model up over full annual cycles to accumulate snow, but I found it confusing to read that Nov-Feb were omitted in paragraph starting around line 160, but previously in the manuscript it was said the albedo would be set to April values during the winter period. So what exactly is being done here?

I also think more description of the MOD10A1 product is needed here. Is it a true daily albedo integrated over a full 24 hour period based on several overpasses of the MODIS instrument, or is there a certain swath that is used for a specific local time? It is unclear what time of day is being used for the optimization. The solar zenith angle varies of course with the day of the year as well as the time of day and thus there is likely a solar zenith angle dependence still in the MODIS albedo product. It is also mentioned that VIS and NIR albedo are used (e.g. Line 70), but my understanding is that the MOD10A1 is a broadband albedo product. Further there are no figures showing VIS and NIR albedo. Since the MOD10A1 data set is being used for the DA, much more information about this data set is required and a discussion of its accuracy. I'm particularly not convinced that there exists a true north/south gradient in the albedo. That doesn't fit with the known pattern of precipitation that brings fresh snow to the ice sheet. The tuned model actually seems to do a better job of expected albedo pattern (e.g. Figure 1). The MODIS pattern looks to be a solar zenith angle dependence. Is the albedo normalized by the Solar Zenith Angle? I also find the MODIS albedo product to show too high of albedos (i.e. Figure 1, Figure 4).

More specific comments.

1. Line 68, do you have a reference to support the statement that Greenland soil type is loam?
2. Lines 90-97 – yes the MODIS retrieval has larger errors under high solar zenith angles (SZA), but most of the ice sheet is dark in winter and thus, there is not albedo retrieved. It's not necessarily that MODIS retrievals are inaccurate, the ice sheet has no sun. Thus, this section needs to be rewritten to be more exact, and also some value of SZA for which you think the MODIS data are inaccurate needs to be stated (along with the appropriate reference).
3. Figure 1. Over which months is the albedo shown? Also, if Figure 1 is averaged from March to October like the other figures are, then the spatial pattern doesn't really make sense to me for the observations which makes me think there is a bias in the

observations. You would expect higher albedo values over the high elevation regions, not a north to south gradient as precipitation patterns do not show this north-south gradient. Future, the albedo values are too high from the observations considering this is summer albedo and the surface is melting over large parts of the ice sheet (see for example melt patterns from passive microwave https://nsidc.org/greenland-today/. For example, I'm including a figure here of the melt in 2022 and thus, you would expect an albedo pattern that loosely follows the microwave melt detection.

[Figure]

4. Figure 2 and its discussion on lines 180-185, I don't follow how you can say you see a degradation in model-data for March to October and that the improvement was only in the winter month. How is that shown? All the images in that figure are stated to be averaged from March to October, so where does one see that there was an improvement only in winter (and at a time when the satellite is not even recording albedo?)
5. Line 185-186, I'm not sure what is meant by that statement. I also do not know which figure the paragraph that follows refers to.

6. Line 212-214, there is no observed albedo in the winter months, so how can you talk about fitting to observed values during winter? Even in mid-January very little of the ice sheet is illuminated. Thus, much more discussion on what is meant by winter and the fitting is needed. I do not necessarily believe that filling in winter values with April values is accurate as the albedo will vary strongly as a function of precipitation and this is completely neglected if you are replacing winter albedo values with those in April.
7. Figure 4, generally new snow has an albedo around 0.85 and thus it is clear that the MOD10A1 values are too high. How does this impact your results and should you really be fitting to something that is unrealistically high?
8. Figure 7. I'm not sure which refer to edge vs. middle as there is no hatching shown in the actual figure.

---

## Author Comment (AC1)

**Reviewer 1**

This paper deals with improving how surface albedo is represented in land surface models by data assimilation of satellite-derived albedo. I will note that I am not an expert on data assimilation and thus, cannot really comment on the applicability of the DA approach used here. The only comment I have about the DA is I'm not sure how all the parameters in Table 1 are tuned to fit the albedo, and maybe some more explanation as to how that is done is warranted. I find the sensitivity analysis ok so no strong comments there. My comments mostly pertain to the MODIS data and result of clarify in the manuscript.

We thank the reviewer for taking the time to read and comment on the manuscript. We welcome comments from someone who is not an expert in DA since we hope to reach a wide audience with this study. This study uses data assimilation for parameter estimation instead of the more common state estimation approach. In state estimation, the model state is updated whilst keeping the model parameters fixed (see paragraph starting on L26 for a list of state estimation studies). For parameter estimation, the parameters are allowed to vary in the ranges prescribed. They are optimised together in this parameter space. The cost function is dependent on the parameters (i.e., Eq. 1: J(x), where x is a vector containing all the parameters). The cost function has two parts, one measuring the mismatch between the model outputs and the observations, and the second part the mismatch between the new parameter values and the original values. Since the model outputs depend on the parameters, changing the parameters changes the model outputs impacting the mismatch between it and the observations (in our case, satellite retrievals). Even parameters not directly involved in the calculation of the snow albedo will impact the output, as we see in the sensitivity analysis, since many processes are interlinked. For example, the ice albedo is not part of the snow albedo equations but impacts the overall albedo calculated.

We have added the following to the manuscript to clarify this:
> "To minimise the cost function, two algorithms were considered in this study. *They both work by varying the full set of parameters considered within the prescribed ranges, retaining the set of parameters at each iteration which reduces J(x) compared to the previous iteration.* The first *algorithm* is a deterministic gradient-based method… "

One general comment, it is true that under low sun angles, the albedo retrieved from satellite is less accurate, yet I find it strange to have a discussion on winter months when the sun is below the horizon. I realize that there is a need to spin the model up over full annual cycles to accumulate snow, but I found it confusing to read that Nov-Feb were omitted in paragraph starting around line 160, but previously in the manuscript it was said the albedo would be set to April values during the winter period. So what exactly is being done here?

We apologise for the confusion caused. The step of setting the winter values to the April values was not undertaken by us but rather by the Box/MOD10A1 product creators. We have clarified in the text which post-processing steps were done by the product generators and which were done by us. Full changes to the paragraph can be found in response to the next comment.

I also think more description of the MOD10A1 product is needed here. Is it a true daily albedo integrated over a full 24 hour period based on several overpasses of the MODIS instrument, or is there a certain swath that is used for a specific local time? It is unclear what time of day is being used for the optimization. The solar zenith angle varies of course with the day of the year as well as the time of day and thus there is likely a solar zenith angle dependence still in the MODIS albedo product. It is also mentioned that VIS and NIR albedo are used (e.g. Line 70), but my understanding is that the MOD10A1 is a broadband albedo product. Further there are no figures showing VIS and NIR albedo. Since the MOD10A1 data set is being used for the DA, much more information about this data set is required and a discussion of its accuracy. I'm particularly not convinced that there exists a true north/south gradient in the albedo. That doesn't fit with the known pattern of precipitation that brings fresh snow to the ice sheet. The tuned model actually seems to do a better job of expected albedo pattern (e.g. Figure 1). The MODIS pattern looks to be a solar zenith angle dependence. Is the albedo normalized by the Solar Zenith Angle? I also find the MODIS albedo product to show too high of albedos (i.e. Figure 1, Figure 4).

We agree with the reviewer that for DA it is important to understand the data set used in the assimilation and its limitations. As such, the section describing the product has been expanded as well as the discussion (see below). Firstly, let us address the different points highlighted in the comment:

- The dataset uses data from the MODIS sensor on board the NASA Terra satellite. The Terra satellite has a sun-synchronous, near-polar circular orbit which crosses the equator at ~10:30 A.M. local time (Hall & Riggs, 2016). Complete global coverage occurs every 1-2 days, with more frequent retrievals occurring near the poles. MOD10A1 is a clear-sky product, and when more than one observation is available, the value representing the best sensor view of the surface in the cell based on solar elevation, distance from nadir, and cell coverage is kept. In this study, we use the data from Box (2017), which is based on the MOD10A1 data but with additional denoising, smoothing and gap-filling. This dataset was specifically created for applications over Greenland (Box et al., 2017) and validated against ground-based measurements from the PROMICE stations (Fausto et al., 2021).

- MODIS has a range of albedo products integrating over different spectral bands e.g., VIS/NIR/SWIR/complete products. The reviewer is correct in stating that the MODIS product used in MOD10A1 is a broadband albedo product. However, in the ORCHIDEE model, VIS and NIR albedo are computed separately. Therefore, to be able to compare the model to the satellite retrievals, we averaged the VIS and NIR albedo model outputs. This is stated on L70 of the ORCHIDEE description section.

- As for solar zenith angle dependence, this is managed first in the MOD10A1 distribution and second in the Box (2017) distribution. For MOD10A1, pixels with solar zenith angles > 70° are masked (where night is defined as a solar zenith angle ≥ 85°). This and other adjustments to the dataset eliminate a spurious darkening trend concentrated over snow and in the northern part of Greenland (Polashenski et al., 2015). Over Greenland, a

residual bias based on the sun's angle above the horizon is observed in MOD10A1 when compared to the in situ measurements. Box (2017)'s dataset corrects this bias according to time and latitude using a linear regression (see Fig. 4 in Box et al., 2017). Furthermore, in Box et al. (2017), Figure 2 shows albedo values measured at an in situ site in Greenland (KPC_L station from the PROMICE network). These values go as high as 0.9 and correlate well with the de-noised MOD10A1 product.

[Figure]

Fig. 2. Year 2013 example comparison of daily de-noised albedo from satellite (NASA MODIS MOD10A1 Collection 6 data) and the ground (PROMICE) for the KPC_L (Fig. 1) station on the north-eastern Greenland ice sheet.

Some SZA bias may remain in the dataset, and we have added lines to acknowledge this in the conclusion. However, it is also likely that the north-south gradient in albedo exists. Although there is more precipitation in the south of the GrIS, fresh snow in the south decays faster due to warmer temperatures. In addition, to melt processes, there are also other processes that impact the GrIS albedo, such as dust deposition and algae growth. For example, on the bare ice zone of the southwestern portion of the Greenland ice sheet, ice algae can account for 75% of albedo variability (Cook et al., 2020; Williamson et al., 2020).

In light of all these comments and responses, the following text has been expanded in the MODIS description section in Methods and Data:

"In this study, we used satellite-derived *snow* albedo from the NASA (National Aeronautics and Space Administration) MODIS (Moderate-Resolution Imaging Spectroradiometer) MOD10A1 product (Hall et al., 1995). *This product uses data from the Terra satellite, which has a sun-synchronous, near-polar circular orbit crossing the equator at approximately 10:30 A.M. local time (Hall & Riggs, 2016) and providing global coverage every 1-2 days. MOD10A1 is a clear-sky daily product. When more than one retrieval is available on a given day, which is the case near the poles, the best value is kept. This best value is chosen based on solar elevation, distance from nadir and cell coverage (Hall & Riggs, 2016). In addition, pixels in the MOD10A1 with solar zenith angles greater than 70° are masked (night is defined as a solar zenith angle greater than 85°).*

*The version of MOD10A1 we used in this study was further processed by Box et al. (2017). Using data from* collection 6 of MOD10A1 (Riggs et al., 2015; Hall and Riggs, 2016), Box et al. (2017) de-noised, gap-filled and calibrated the data into a daily 5km

grid covering Greenland for the years 2000-2017. *This dataset was further validated against ground-based measurements from the PROMICE stations (Fausto et al., 2021) and the residual bias in the dataset based on the* solar zenith angle *corrected for using a linear regression according to time and latitude (Box et al., 2017). Finally, in this dataset,* when  *there is* not enough solar illumination *to compute the albedo* during the winter months (January, February, November, and December), *Box et al. (2017)'s distribution swaps in the* April values .

In this study, we *used the dataset created by Box et al. (2017),* further aggregating these data to the resolution of the ORCHIDEE outputs, imposed by the meteorological forcing files (20 km)."

and to the discussion and conclusions:

"We must also remember that there are errors linked to the satellite retrievals themselves. Indeed, the large uncertainties in the winter months led us to omit them  for this study. For the other months, we set the observation errors to be the mean-squared difference between the observations and the prior model simulation to also account for the structural model errors. However, in practice, the true errors may be very different. *For example, although steps to correct the solar zenith angle bias in the product have been untaken, it is possible that the strength of the north-south albedo gradient observed in the data is an artefact of the product. Without clear and robust uncertainty quantification, we cannot disentangle natural GrIS processes from biases in the retrievals. There is an urgent need for data producers to provide this uncertainty, ideally at each time step.*"

More specific comments.

1. Line 68, do you have a reference to support the statement that Greenland soil type is loam?
   Citation to the HWSD global soil texture map has been added.

2. Lines 90-97 – yes the MODIS retrieval has larger errors under high solar zenith angles (SZA), but most of the ice sheet is dark in winter and thus, there is not albedo retrieved. It's not necessarily that MODIS retrievals are inaccurate, the ice sheet has no sun. Thus, this section needs to be rewritten to be more exact, and also some value of SZA for which you think the MODIS data are inaccurate needs to be stated (along with the appropriate reference).
   We thank the reviewer for these precisions. We agree that referring to the retrievals as inaccurate is not strictly correct. We have rewritten the MODIS description section to be more precise and included all the comments above about the product specifics and limitations (see above for the new paragraph in response to earlier comments).

3. Figure 1. Over which months is the albedo shown? Also, if Figure 1 is averaged from March to October like the other figures are, then the spatial pattern doesn't really make

sense to me for the observations which makes me think there is a bias in the observations. You would expect higher albedo values over the high elevation regions, not a north to south gradient as precipitation patterns do not show this north-south gradient. Future, the albedo values are too high from the observations considering this is summer albedo and the surface is melting over large parts of the ice sheet (see for example melt patterns from passive microwave https://nsidc.org/greenland-today/. For example, I'm including a figure here of the melt in 2022 and thus, you would expect an albedo pattern that loosely follows the microwave melt detection.

Figure 1 showed all months. We have replaced these with ones averaged over just the summer months for consistency. The albedo values are lower now that the figures consider only the summer albedo and this has been clarified in the caption:

"Retrieved and simulated mean albedo over Greenland (averaged over March-October for 2000-2017)"

4. Figure 2 and its discussion on lines 180-185, I don't follow how you can say you see a degradation in model-data for March to October and that the improvement was only in the winter month. How is that shown? All the images in that figure are stated to be averaged from March to October, so where does one see that there was an improvement only in winter (and at a time when the satellite is not even recording albedo?)

These results were part of the preliminary experiments used to find the ideal setup for the main experiment (see L156). These experiments helped to a) choose the optimisation algorithm to use in the main experiment, b) pick the time period required and c) the weightings required. It was during these experiments that we realised how much these winter months impacted the optimisation and decided to exclude them from the main experiment since they are not "true" observations. However, to be consistent with the rest of the manuscript, we have redone these optimisations over Mar-Oct only and included these instead.

5. Line 185-186, I'm not sure what is meant by that statement. I also do not know which figure the paragraph that follows refers to.

We acknowledge that this statement about local minima comes a bit out of left field for people unfamiliar with DA. To find the best set of parameters, we are minimising the cost function (Eq 1) with respect to the parameters. If we think about our cost function as a simple curve (i.e., J(x) on the y-axis and parameter x on the x-axis), within the range allowed for the parameter, there will be a minimum value (where the gradient is zero and positive on either side). However, if the curve is not a smooth bowl in the given range but rather an undulating function, this will not necessarily be the only place where the gradient is zero with positive values on either side. All examples where this is true are called "local" minima, and the "global" minimum is the minimum of all these. The gradient-based algorithm uses the negative gradient values to find how to reduce the cost function. If there are only positive gradients around a point, then the gradient-based algorithm becomes stuck. We are at a minimum but not necessarily the global minimum. When manually tuning a model, the same can be true - we only change a parameter

manually when the RMSD decreases, the same as a negative gradient. We have added the following text:

> "Since the prior model used was already extensively manually tuned, it is likely that we started very near to a minimum *(i.e., somewhere where the gradient is close to zero surrounded by positive gradient values). However, this is not the global minimum since we have been able to reduce the cost function further when using a different algorithm (i.e., in the GA case). Since gradient-based algorithms rely on negative gradient values to minimise the cost function,*  the gradient-based algorithm is unable to leave local minima, *and therefore,* the cost function is hardly minimised."

For the following paragraph, we have added a reference to Figure 2 to the text as well as titles to each panel of the figure for clarification.

6. Line 212-214, there is no observed albedo in the winter months, so how can you talk about fitting to observed values during winter? Even in mid-January very little of the ice sheet is illuminated. Thus, much more discussion on what is meant by winter and the fitting is needed. I do not necessarily believe that filling in winter values with April values is accurate as the albedo will vary strongly as a function of precipitation and this is completely neglected if you are replacing winter albedo values with those in April.

The reviewer is correct that we need to be more precise with the language used and not talk about fitting the albedo in winter. To simplify matters, we have removed the winter months from the analysis altogether. Again, we stress that setting the winter values with April was not done by us. This is a feature of the product used. Nevertheless, by removing the winter months from the analysis, we should avoid dealing with these issues.

7. Figure 4, generally new snow has an albedo around 0.85 and thus it is clear that the MOD10A1 values are too high. How does this impact your results and should you really be fitting to something that is unrealistically high?

In the optimisations, we include uncertainty in the cost function (matrix R, see L109 and Discussion). This term helps stop overfitting and takes these types of errors into account. In future works, we could consider having a variable uncertainty to take into account the larger uncertainty during winter. However, the Box distribution of the MOD10A1 collection 6 product does not have uncertainties included, so it would be hard to accurately determine an uncertainty time series to use at each pixel. This is an ongoing issue with remote sensing and land surface modelling - it is extremely hard to quantify these uncertainties. When uncertainties are provided with remote sensed products, they tend to be uncertainties due to the sensors or retrieval algorithm used. Data users need to also think about the uncertainty around the representativity of ground-based measurements and different resolutions. As discussed in the conclusions, we set the uncertainty to be the mean-squared difference between the observations and the prior model simulation. This tends to be quite a conservative way to set these uncertainties and is widely used in data assimilation studies. Nevertheless, we have expanded the

discussion to highlight the importance of these uncertainties and as a recommendation to data providers:

> "*Without clear and robust uncertainty quantification, we cannot disentangle natural GrIS processes from biases in the retrievals. There is an urgent need for data producers to provide this uncertainty, ideally at each time step.*"

8. Figure 7. I'm not sure which refer to edge vs. middle as there is no hatching shown in the actual figure.
The caption of the figure has been expanded to contain a clearer explanation of how to distinguish between the middle and the edge sensitivities, and the hatching in the legend changed to be filled in instead.

> "In each case, the sensitivity of the parameters is shown for simulated quantities at the edge of the ice sheet *(shown by the filling at the edge of each box)* and in the middle of the ice sheet *(shown by the filling in the middle of each box)*."

**References**

Box, J. E., Van As, D., and Steffen, K.: Greenland, Canadian and Icelandic land-ice albedo grids (2000–2016), GEUS Bulletin, 38, 53–56, 2017

Cook, J. M., Tedstone, A. J., Williamson, C., McCutcheon, J., Hodson, A. J., Dayal, A., Skiles, M., Hofer, S., Bryant, R., McAree, O., et al.: Glacier algae accelerate melt rates on the south-western Greenland Ice Sheet, The Cryosphere, 14, 309–330, 2020

Fausto, R. S., van As, D., Mankoff, K. D., Vandecrux, B., Citterio, M., Ahlstrøm, A. P., Andersen, S. B., Colgan, W., Karlsson, N. B., Kjeldsen, K. K., et al.: Programme for Monitoring of the Greenland Ice Sheet (PROMICE) automatic weather station data, Earth System Science Data, 13, 3819–3845, 2021.

Hall, D. and Riggs, G.: MODIS/Terra Snow Cover Daily L3 Global 500m Grid, Version 6. Greenland coverage., National Snow and Ice Data Center, NASA Distributed Active Archive Center, Boulder, Colorado USA., http://nsidc.org/data/MOD10A1/versions/6, accessed December 2016., 2016.

Hall, D. K., Riggs, G. A., and Salomonson, V. V.: Development of methods for mapping global snow cover using moderate resolution imaging spectroradiometer data, Remote sensing of Environment, 54, 127–140, 1995

Polashenski, C. M., J. E. Dibb, M. G. Flanner, J. Y. Chen, Z. R. Courville, A. M. Lai, J. J. Schauer, M. M. Shafer, and M. Bergin, Neither dust nor black carbon causing apparent albedo decline in Greenland's dry snow zone: Implications for MODIS C5 surface reflectance, Geophys. Res. Lett., 42, 9319–9327, 2015

Williamson, C.J., Cook, J., Tedstone, A., Yallop, M., McCutcheon, J., Poniecka, E., Campbell, D., Irvine-Fynn, T., McQuaid, J., Tranter, M. and Perkins, R. Algal photophysiology drives

darkening and melt of the Greenland Ice Sheet. *Proceedings of the National Academy of Sciences*, *117*(11), pp.5694-5705, 2020

---

## Author Comment (AC2)

**Reviewer 2**

This article presents the calibration of the ORCHIDEE model against the MODIS derived snow albedo dataset. While the overall objective of improving albedo is very relevant, this particular study, in my opinion, is very limited. My specific concerns are outlined below.

We thank the reviewer for taking the time to read and comment on the manuscript. We believe addressing these comments will help widen the scope of the paper, especially by adding a section on the impact of different parameters. Although this study is an example over Greenland, the techniques used and model developments will be relevant to other applications. This study helps show how satellite data is vital for model development. We improve the model-data fit through robust Bayesian parameter calibration and identify further model developments needed to capture the GrIS processes fully. In ice sheet modelling, this type of robust Bayesian parameter estimation is rare, and its value in identifying missing processes is not well documented. Furthermore, we use the full area for parameter estimation (compared to using selected pixels), whereas passed examples focus on only a handful of in situ sites.

We have added the following text to the end of the Introduction to emphasise this:
> "Using MODIS snow albedo, in this study, we use DA for parameter estimation to improve the albedo parameterisation inside the ORCHIDEE LSM (Krinner et al., 2015). Instead of using a single or multisite approach which samples the space, here, to exploit the full spatial coverage of the satellite retrievals, we optimise over the whole area of the GrIS to obtain one best set of model parameters applicable over the full ice sheet. *Although this study is only over the GrIS, we can apply the method to other regions. We show how robust Bayesian parameter estimation is an important tool for model development. We further highlight the different limitations and considerations needed to apply such an approach.*"

The article reads like a description of the research in the way it was conducted. The authors describe all the methodologies the authors tried, which are sometimes distracting from the main objective of the paper. For example, Section 3.2. describes the results with two different optimization algorithms. As shown here (and as well known), gradient search methods have limitations in exploring complex decision spaces within an optimization context. The results presented here are not adding anything new to the key focus of this paper, and it is distracting. In Section 2.4.2 – It is not clear (at this point) in the manuscript what is meant by 'performing a sensitivity analysis of the model'. Typically, this is done ahead of the calibration step to reduce the number of parameters being optimized (as the authors acknowledge in Section 3.4.3). If that's the same context, it'll be good to describe that and present this section before 2.4.1. Similarly, Section 3.4.3 should be presented earlier (even though that's not how this work evolved). I appreciate the value of explaining all the steps, but there are lots of 'preliminary' setups (section 3.2, line 207) in this paper. A major recommendation is to restructure the paper so that it focuses on the finalized results, while presenting the intermediate results and steps only to support the main findings.

We thank the reviewer for their suggestion to restructure the paper. We agree that this will help widen the focus of the study. As suggested, we have moved the sensitivity analysis to the beginning of the study - as well as clarified in Section 2.4.2 what is meant by performing an SA.
> "*An SA tests the sensitivity of a model output (usually a physical variable). It tests how the output changes, with respect to different inputs - here the model parameters.*"

The defining edges section, which is needed before the SA, has been moved ahead of the "Performed experiments" section in "Methods and Data". "Defining edges" and "Performed experiments" are now under the umbrella of "Experimental setup". Although we believe that for the non-DA expert, we should keep the analysis of the different methods, this has been moved to the Appendix so as to not distract from the main results. Similarly, we have also moved the edge-only optimisation to the Appendix. Both sections are now under the "Preliminary optimisations" heading. To account for these changes, Sect. 2.5 (Performed experiments) and the Discussion section have been rewritten.

As the authors note in the summary, calibration has its problems in that adjusting certain model parameters may improve some parts of the model, while degrading others. The main objective of improving albedo is to improve the changes in the snow pack over GrIS, as noted in the intro. The paper needs to describe what the impact of the improved snow albedo formulation is on the snow simulations (and other model states). Does the improved albedo lead to better snow states?

Yes, we agree that this information is missing; as such, we have added the following section to discuss the impact on the rest of the snow states:

> "### 3.3.3 Impact of the different parameter sets on modelling the surface mass balance of the Greenland Ice Sheet
>
> *In Fig. 7, we consider how the different parameter sets discussed in this study impact the modelled snow states. To assess the performance of the different ORCHIDEE parameter sets, we compare the model outputs to that of the MAR model. Although MAR is a model with its own biases and errors, it has been shown to have good estimations of the different snow states (Fettweis et al., 2017, 2020) and so is a good product against which to compare.*
>
> *In particular, we are interested in better modelling the surface mass balance (SMB). It measures the difference between mass gains and ablation processes, hence dominating the rates of mass change over the GrIS. Compared to MAR, the manually tuned version of ORCHIDEE performs best at simulating SMB. This can be seen both spatially and temporally. Spatially, the differences between MAR and the ORCHIDEE simulations are observed at the edges - especially in the north and west of the GrIS. The most noticeable difference in the ORCHIDEE runs can be seen at the west of the ice sheet, where the tuned model simulates SMB the best when compared to MAR, followed by the optimised model. In both the manually tuned and optimised models, the SMB is reduced at the west of the ice sheet compared to the default ORCHIDEE model. This is mirrored by an increase in runoff at the west of the ice sheet. Indeed, for simulated runoff, changes are mainly found at the west of the ice sheet, with the tuned model performing the best and the optimised model second when compared to MAR. Both models improve the fit compared to the default ORCHIDEE simulations. However, neither model is able to capture the magnitude of the runoff in summer, with the tuned model still only simulating half the expected magnitude of runoff.*
>
> *When we consider modelled sublimation, we get the most different results. By increasing the albedo over the ice sheet, we decrease latent heat over the area and hence sublimation. When considering the time series, we see that the optimised model gets the correct magnitude of sublimation during the summer months. All of the ORCHIDEE simulations have a delayed peak compared to MAR and no sublimation is simulated by*

[Figure]

Figure 7: Impact of different parameter sets on ORCHIDEE simulations; "Standard" uses default parameter values, "Tuned" uses parameter values from the manual tuning and "Optimised" from the ORCHIDAS optimisation. The top panels show spatial maps averaged over time (March-October) of MAR (left) and ORCHIDEE-MAR; the bottom panels show monthly means averaged over space.

*ORCHIDEE outside the summer months. When averaged over time, we see that MAR has high sublimation rates to the east of the GrIS. However, none of the ORCHIDEE simulations capture this. Instead, the sublimation over the centre of the ice sheet is what changes with the different parameter sets - with the optimised model lowering the rates the most. The strong impact that changing albedo has on simulated sublimation over the whole of the GrIS shows how coupled they are in the model.*

*Overall, with the optimised model, we do better than the standard ORCHIDEE model but not as well as the tuned model. During the manual tuning of the albedo parameters, the performance of the new parameters was assessed against several model outputs, including SMB, sublimation and runoff at each step of the trial and error procedure. We can think of this manual tuning as a multi-objective calibration. When performing the Bayesian optimisation, we get the best fit to the albedo. However, we overfitted to albedo with no other data, degrading the fit to other model outputs. As seen with the BFGS algorithm and the posterior parameters, parameter space is not smooth but has many local minima. As such, it is possible that a different solution exists, reducing the albedo to a similar extent whilst also improving the fit to other modelled outputs. To achieve this, we need to include more data in the optimisation to perform a multi-objective optimisation. If we cannot find such a parameter set, this would point to structural problems in the model, i.e., missing processes.*

We have also added the following to the discussion:

*"We also showed the influence of the parameters on other model outputs by comparing simulated snow states to the MAR model. The optimised model was found to perform better than the original ORCHIDEE model but not as well as the tuned model for simulating SMB and runoff. For sublimation, the optimised model simulated the most accurate magnitude in summer; however, it still showed a bias when considered spatially."*

The modeling setups use forcing data from MAR, which is a modeled estimate, presumably with its own associated biases and errors. In a calibration setup, the tuned parameters are then used as an error sink to 'hide' these boundary condition errors. This needs to be discussed in the article. Is there an evaluation of MAR data over GrIS? Are other 'observational' datasets available?

Yes, the reviewer is right to highlight this. Although we mention in the Conclusions that using a different meteorological forcing would help separate model structural errors from errors in the forcings, we do not explain that the data assimilation might correct bias in the data instead of errors in the land-surface model. This is made more explicit in the Conclusions:

> "Since we are running  ORCHIDEE offline - i.e., prescribing the meteorological forcing - it would also be beneficial to run the model with different forcings to separate model structural errors from the errors in the forcing. *This is important since MAR is a modelled estimate and, therefore, will be subject to its own biases and errors. We would want to ensure that we are correcting errors in the land surface model and not correcting atmospheric biases in the forcing data.*"

There are two stand-out studies evaluating MAR over GrIS. The first is Fettweis et al. (2017), where different forcings are tested. The second is Fettweis et al. (2020), where MAR was found to be one of the best models for simulating SMB in an intercomparison study. Since we also now use MAR to evaluate the different snow states of the optimised model, we have added these references to that section.

MAR is a regional atmospheric model that uses forcing data to prescribe the atmospheric boundary conditions outside the domain. Here ERA-Interim data is used, the predecessor of ERA5. This has been added to the MAR description in Sect. 2.2:

> The ORCHIDEE model was forced using meteorological outputs from the regional climate model Modèle Atmosphérique Régional (MAR; Gallée and Schayes (1994)) *version 3.11.4. MAR is a regional atmospheric model that uses 6 hourly ERA-Interim reanalyses data from the European Centre for Medium-Range Weather Forecasts (ECMWF, Dee et al., 2011) to prescribe the atmospheric boundary conditions outside the domain.* Outputs from the MAR have a resolution of 20 km *and a 3 hourly time step. In addition to the MAR meteorological outputs, we consider* runoff, sublimation and SMB outputs in this study to assess the impact of the optimisation on these simulated quantities.

We could consider using global reanalysis products to run the model e.g., ERA5. However, as global products instead of regional ones, most of these products have a coarser spatial resolution and are less understood over Greenland. As for observational datasets with which to evaluate the model, we have already listed a few in the conclusion, e.g. GRACE, LST_cci. However, evaluating against these data is outside the scope of this study.

Since ORCHIDEE is used in global setups, how are the results over this domain applicable in a general sense? Are these calibrated parameters limited to GrIS?

Ongoing work is looking at other regions to test the applicability of calibrated parameters. Although this is out of the scope of this paper, we have added the following to the conclusion to highlight this perspective:

> "For additional evaluation, we are testing the application of this model and parameters to other polar and non-polar regions, starting with other ice sheets such as Antarctica."

**Minor comments:**

Line 51. Need brackets around Krinner et al. (2005).
Done

Line 67: Change to 'the' instead of 'our'?
Done

Section 2.3 – It is important to clarify (here, early on in the paper, abstract, and title) that the snow albedo is being calibrated instead of the total albedo. MODIS has several different albedo products (blue-sky, black-sky etc.) Please clarify.
Yes, we agree with the reviewer that we need to clarify which product is used, given that MODIS has several. We have therefore added "snow albedo" to the title, abstract, introduction etc. as suggested. We have also expanded the MODIS description section (in line with comments from Reviewer 1):

"In this study, we used satellite-derived *snow* albedo from the NASA (National Aeronautics and Space Administration) MODIS (Moderate-Resolution Imaging Spectroradiometer) MOD10A1 product (Hall et al., 1995). *This product uses data from the Terra satellite, which has a sun-synchronous, near-polar circular orbit crossing the equator at approximately 10:30 A.M. local time (Hall & Riggs, 2016) and providing global coverage every 1-2 days. MOD10A1 is a clear-sky daily product. When more than one retrieval is available on a given day, which is the case near the poles, the best value is kept. This best value is chosen based on solar elevation, distance from nadir and cell coverage (Hall & Riggs, 2016). In addition, pixels in the MOD10A1 with solar zenith angles greater than 70° are masked (night is defined as a solar zenith angle greater than 85°).*

*The version of MOD10A1 we used in this study was further processed by Box et al. (2017). Using data from* collection 6 of MOD10A1 (Riggs et al., 2015; Hall and Riggs, 2016), Box et al. (2017) de-noised, gap-filled and calibrated the data into a daily 5km grid covering Greenland for the years 2000-2017. *This dataset was further validated against ground-based measurements from the PROMICE stations (Fausto et al., 2021) and the residual bias in the dataset based on the* solar zenith angle *corrected for using a linear regression according to time and latitude (Box et al., 2017). Finally, in this dataset,* when  *there is* not enough solar illumination *to compute the albedo* during the winter months (January, February, November, and December), *Box et al. (2017)'s distribution swaps in the* April values .

In this study, we *used the dataset created by Box et al. (2017),* further aggregating these data to the resolution of the ORCHIDEE outputs, imposed by the meteorological forcing files (20 km)."

Line 146: change to 'output' instead of 'writing'
Done

Line 151: remove 'However'

Done

How do the calibrated values influence the peak winter month simulations?
Since the retrievals in the winter months are uncertain, we have decided to remove all analysis of model-data fit to these months.

Figure 1 – this is the snow covered albedo? Is this average computed by excluding Nov-Feb?
This is retrieved snow albedo against simulated albedo. However, this can be ice albedo in areas where the snow has melted. This figure was computed over the whole year. However, this has now been changed to exclude Nov-Feb for consistency with the rest of the manuscript.

Section 3.1 – This is a very hand-wavy section. The authors need to spell out exactly what was changed in this manual calibration procedure. What parameters/physics were changed?
The parameter values in Table 1 were the values resulting from the manual tuning experiments. We have replaced the values with the default ORCHIDEE values, clarifying this in the caption of the Table. In Section 3.1, we explain how the parameters have been changed from these default values. We have added a table to the Appendix showing the resulting tuned values. This table also shows the posterior parameter values from the optimisation for completeness.

Below is the text we have added to Sect 3.1 to explain how the parameters were changed in the tuning experiment:
> "Before using ORCHIDAS to optimise the model parameters, the ORCHIDEE model was first tuned manually *through trial and error*. While not as robust as using a Bayesian framework, this initial step is common for land surface modellers and helps get a sense of the different parameter sensitivities. The primary focus of this manual tuning was to better capture the behaviour of the GrIS at its edges. *This was achieved by increasing the overall albedo of fresh snow ($A_{aged}$ + $B_{dec}$) and the snowfall depth required to reset the snow age ($\delta_c$), while also decreasing the albedo of aged snow and decreasing the rate of snow age decay ($\tau_{dec}$). Furthermore, one of the tuning constants for glaciated snow-covered areas was decreased ($\omega$). The rest of the parameters were kept as the default ORCHIDEE parameters (see Table B1 for full results).*
>
> This initial tuning helped the model to better simulate the albedo at the edges of the ice sheet, especially in the western part (Fig. 2)*, as well as other snow states such as SMB and runoff, which were also used to assess the success of the manual tuning*."

Section 3.2: How many iterations of GA were used here? Are these the results from the 'Both' approach (results in Figure 2)?
We used 15 iterations for the GA algorithm, which we found to be sufficient for convergence. This information has been added to the "Performed experiments" section (Sect. 2.5):
> "*We found that the genetic algorithm greatly outperformed the BFGS algorithm, reducing the cost function by 11% compared to a negligible reduction, and that 15 iterations of the genetic algorithm were sufficient for convergence.*"

Are these the results from the 'Both' approach (results in Figure 2)?
This is an optimisation over the whole ice sheet without any weighting. This section was about finding the right method. It led us to identify that the middle points dominated the optimisation, necessitating weighted edges in the main optimisation. This is now made explicit in the Preliminary experiments section:
> "*and over the whole of the GrIS (without weighting the edges).*"

Table 2: How are the albedo evaluated for 'All months'? If you don't trust the MODIS albedo during the winter months, how do you justify comparing back to them?

We agree that assessing the fit to the winter months is tricky and not precise, given the high uncertainties during these periods. We have removed all analyses using the winter months. This is also in line with the comments from R1.

Line 220: Why were these three years chosen? How do you do these calibrations (separately for each year and somehow harmonize the calibrated parameters? Or are they calibrated from a single run, but the calibration data is withheld during all years except 2000, 2010, and 2012)?

As stated in the "Performed optimisations" section, these years were randomly selected. In section (L159-162), we also explain that the three years were optimised simultaneously. This means that the cost function is a sum of three cost functions, one for each year considered. This approach was chosen to reduce computation costs since even running ORCHIDEE over the GrIS for one year at this resolution takes 20 minutes of computation time on our computing cluster run in parallel over eight cores. We have expanded the text in the "Performed optimisations" to be more explicit:

> "We optimised over these three years simultaneously. *This means that, in this main experiment, we minimised a cost function comprising a sum of three cost functions, one for each year considered.*  The rest of the 2000-2017 time series was used for validation."

and added to the words "randomly selected", "simultaneously", and a reference to Sect. 2.5.2 in L220:

> "For the main optimisation, the GrIS albedo was optimised over the *randomly selected* years 2000, 2010 and 2012 *simultaneously*, with a larger weight given to the edges *(see Sect. 2.5.2 for the full setup description)*."

Line 223: Add a comma after 'Indeed'.

Done

Line 232: Why is it that 'We would not expect to lower the RMSD of the edges any further'?

Since we used the genetic algorithm in the edge-only optimisation, we can say with some confidence that the algorithm converged at a global minimum. We found the best set of parameters given the setup and, more importantly, the lowest cost function value we could achieve. Since the main optimisation included more data to fit, it was possible that the algorithm would reduce the cost function by fitting the other data (i.e., the middle) and not the edges since the middle has three times more data points. We have rephrased the text to be more explicit:

> "However, the RMSD reductions over the edge points are similar in magnitude to the reductions found in the preliminary optimisation where only the edge points were considered (Table A1). This means that the weighting used between the edge and middle points during the optimisation was sufficient - *we have achieved as low RMSD at the edges as in the edge-only experiment*. "

**References**

Box, J. E., Van As, D., and Steffen, K.: Greenland, Canadian and Icelandic land-ice albedo grids (2000–2016), GEUS Bulletin, 38, 53–56, 2017

Box, Jason E., 2022, "MODIS Greenland albedo", https://doi.org/10.22008/FK2/6JAQPK, GEUS Dataverse, V1

Hall, D. and Riggs, G.: MODIS/Terra Snow Cover Daily L3 Global 500m Grid, Version 6. Greenland coverage., National Snow and Ice Data Center, NASA Distributed Active Archive Center, Boulder, Colorado USA., http://nsidc.org/data/MOD10A1/versions/6, accessed December 2016., 2016.

Hall, D. K., Riggs, G. A., and Salomonson, V. V.: Development of methods for mapping global snow cover using moderate resolution imaging spectroradiometer data, Remote sensing of Environment, 54, 127–140, 1995

Fausto, R. S., van As, D., Mankoff, K. D., Vandecrux, B., Citterio, M., Ahlstrøm, A. P., Andersen, S. B., Colgan, W., Karlsson, N. B., 420 Kjeldsen, K. K., et al.: Programme for Monitoring of the Greenland Ice Sheet (PROMICE) automatic weather station data, Earth System Science Data, 13, 3819–3845, 2021.

Fettweis, X., Box, J. E., Agosta, C., Amory, C., Kittel, C., Lang, C., van As, D., Machguth, H., and Gallée, H.: Reconstructions of the 1900–2015 Greenland ice sheet surface mass balance using the regional climate MAR model, The Cryosphere, 11, 1015–1033, https://doi.org/10.5194/tc-11-1015-2017, 2017.
Hall, D. K. and G. A. Riggs. 2016. MODIS/Terra Snow Cover Daily L3 Global 500m SIN Grid, Version 6. [Indicate subset used]. Boulder, Colorado USA. NASA National Snow and Ice Data Center Distributed Active Archive Center. https://doi.org/10.5067/MODIS/MOD10A1.006. [Date Accessed].

Stroeve, J. C., Box, J. E., & Haran, T. (2006). Evaluation of the MODIS (MOD10A1) daily snow albedo product over the Greenland ice sheet. Remote Sensing of Environment, 105(2), 155-171. https://doi.org/10.1016/j.rse.2006.06.009

---

## Referee Report (RR1)

[referee-annotated manuscript omitted]

---

## Referee Report (RR2)

Review on "Improving modelled albedo over the Greenland ice sheet through parameter optimisation and MODIS snow albedo retrievals"

**OVERVIEW**

This paper aims to improve simulating snow albedo by the ORCHIDEE land surface model via assimilation of MODIS snow albedo retrievals and parameters calibration through ORCHIDAS. The domain of study is the Greenland Ice Sheet (GrIS). ORCHIDEE is used in offline mode forced using atmospheric data coming from the MAR model. The optimisation is performed over three random years (2000, 2010 and 2012) taken over the period of study 2000-2017 while the improvement of snow albedo is checked over the whole period of study. An increased weight is given to pixels defining the edges of the GrIS compared to other pixels in order to improve albedo simulations over those areas crucial in term of mass loss. Moreover, a comparison between outputs of ORCHIDEE and MAR on surface mass balance, runoff and sublimation is carried out as well as a Morris sensitivity analysis of various parameters related to snow albedo and snow density relative to several outputs such as surface temperature, sensible and latent heat fluxes in addition to the previously mentioned outputs.

**GENERAL COMMENTS**

The paper covers the timely question of snow/ice albedo modelling over the GrIS in the context of increased melt in links with global warming. Here the chosen approach is to assimilate satellite-derived albedo retrievals, which only few studies have performed before. As such, the novelty aspects of the paper are self evident. Therefore, such work would clearly deserve to be published. Nevertheless, I have some reservations about the methodology and I especially find the paper difficult to read. As indicated by one previous referee, *"The article reads like a description of the research in the way it was conducted."*. Unfortunately, after the first revision, this problem still remains. My main issues with the paper are the following:

- The goal of the paper is straightforward: improve snow albedo by assimilation. But why do we want to do that? In which context? Global land surface modelling, if so, why using MAR instead of ERA5 as atmospheric forcing? Or are the targets are ice sheets and large glaciers? During all the several readings I made, I asked myself what were the purposes of such study and I could not find the answer anywhere in the introduction where normally one would expect to find such information.

- The methodology clearly misses validation using independent datasets. Modelled albedos are compared with MODIS data that are partially assimilated (3 years over 18 years). Then ORCHIDEE outputs of surface mass balance, runoff and sublimation are compared with outputs from MAR, while ORCHIDEE is forced using atmospheric data coming from MAR itself. Obviously both model outputs are related, the main difference would come from the modelling approach of snow, ice and albedo in ORCHIDEE and MAR. As such, MAR cannot be considered as a reference as stated by the authors. I do, however, acknowledge that the authors raise some reservations about the limitation of the comparison in the paper.

- Several peripheral considerations (such as the test of the two minimisation approach, L-BFGS-B and genetic algorithm) degrade the readability of the paper while providing very little novelty in terms of science.

As I indicated beforehand, this paper cannot be published in The Cryosphere in its current version. Nevertheless, I do think the authors, who are renowned experts in their fields of expertise, have the

ability to improve the paper to reach the publication stage. It would be unfortunate that such good science would, in the end, not be published. I list my comments and questions section by section below:

**SPECIFIC COMMENTS**

About the objectives of the paper:

- Please specify loudly the context and the purposes of the study in the introduction. Is the goal of the study to focus on ice sheets and large glaciers? If so, why? If not, do the authors focus on global or large scale climate simulations? If so, again why? Please provide also adequate references to justify your choices.

- Justify more adequately the use of MAR in relations to your goals either in the introduction and section 2.2 describing MAR. MAR is indeed a fine atmospheric model well adapted to Greenland but by using MAR, the authors make their approach less replicable to other parts of the world if their goal is to fine tuning parameters for modelled albedo for ice sheets and large glaciers as optimal parameters found are likely to depend on atmospheric inputs.

- The authors starts to talk about runoff, sublimation and surface mass balance (SMB) from L. 92-93. This should be specify in the introduction why those quantities are considered in the discussion section of the paper. Also MAR includes a modelled albedo. How does modelling in ORCHIDEE compare with MAR's? Might be worth comparing both if possible.

- Mass loss in Greenland is not only due to snow melting but also ice dynamics in outlet glaciers that are located in many edge pixels. This should be mention somewhere as it is a limitation of the authors' work (no ice dynamics considered) in the introduction with adequate references such as:
  - Aschwanden, A., Fahnestock, M. & Truffer, M. Complex Greenland outlet glacier flow captured. *Nat Commun* **7**, 10524 (2016). https://doi.org/10.1038/ncomms10524
  - Khan, S.A., Bjørk, A.A., Bamber, J.L. *et al.* Centennial response of Greenland's three largest outlet glaciers. *Nat Commun* **11**, 5718 (2020). https://doi.org/10.1038/s41467-020-19580-5

- Darkening of the GrIS is mentioned in the introduction as a very important phenomenon for albedo. Nevertheless, this darkening is not explored anywhere in the paper. The ability of the authors' approach to capture this darkening could be studied in the result section.

- Since this paper is about albedo assimilation, a longer paragraph dedicated to that very subject would be much appreciated. Also several references are missing, among others:
  - Dumont, M., Durand, Y., Arnaud, Y. and Six, D.: Variational assimilation of albedo in a snowpack model and reconstruction of the spatial mass-balance distribution of an alpine glacier, J. Glaciol., 58(207), 151-164, doi: 10.3189/2012JoG11J163, 2014.

  - Cedelnik, J., Carrer, D., Mahfouf, J.-F. and Roujean, J.-L.: Impact Assessment of Daily Satellite-Derived Surface Albedo in a Limited-Area NWP Model, J. Appl. Meteorol. Clim., 51(10), 1835-1854, https://doi.org/10.1175/JAMC-D-11-0163., 2012.

  - Boussetta, S., Balsamo, G., Dutra, E., Beljaars, A. and Albergel, C.: Assimilation of surface albedo and vegetation states from satellite observations and their impact on numerical weather prediction, Remote Sens. Environ., 163, 111-126, https://doi.org/10.1016/j.rse.2015.03.009, 2015.

- ○ Geppert, G.: Analysis and application of the ensemble Kalman filter for the estimation of bounded quantities, PhD thesis, Universität Hamburg, Hamburg. Doi: 10.17617/2.2161673, 2015.

- A plan at the end of the introduction describing the sections developed in the paper would improve the readability of the paper to a great extent.

About the methodology:

- The set of parameters is optimized over the whole GrIS but albedo conditions varies greatly between the interior and the edges of the GrIS as stated in the paper. Does it make sense to follow this approach instead of a multisite approach? Also, Figure 4b shows differences between Northern and Southern parts of the GrIS.

- The Morris sensitivity analysis does not intervene in the choice of optimized parameters. It should be instead include in a discussion section after the analysis of results. Also it involves parameters that are not optimized. The Morris sensitivity analysis and the comparison between MAR outputs and ORCHIDEE outputs could be merged in a same discussion section as they tend to complete each other. This discussion should also be pointed out in the introduction of the paper. Also, the list of parameters of interest should have be provided in the ORCHIDEE land surface subsection. Since the authors focus not only on albedo per se but also on rate of density change and parameters controlling surface mass balance and runoff, authors should provide more information on the snow model employed in ORCHIDEE or at least point towards adequate references. This would help the readers to understand more clearly the authors' objectives.

- $\tau_{max}$ seems to have almost no influence for albedo according to the Morris sensitivity analysis, why keeping it for optimization?

About MODIS data:

- About MODIS albedos, it is unclear to me if there are any reliable observations during winter time. Authors exclude data from November to February in their data assimilation system but later states albedo is improved in winter times in the result sections. This is rather confusing. I suggest you exclude all albedo comparison for winter period as I do not think they make sense (see e.g. sentence L. 280-281 *"We also see that the optimisations improve the fit … in the optimisation"*). Alternatively, authors can explicit in the manuscript their methodology regarding albedo during winter time.

- How do the authors aggregate MODIS data from the original grid to the MAR grid (just an average or something more sophisticated)? That should be made more explicit in the text.

About the snow and albedo modelling and spin up:

- L. 69-71: *"For the new icy soil type, … those of the loam soil type because it is the dominant soil type in the non-ice-free reguibs around the GrIS (Fischer et al., 2008)"*. Does it make sense to assume that basically, the icy soil type is impermeable (porosity and saturation are equal) for the edge of ice sheets? What is it classically used for ORCHIDEE in CMIP experiments and how do your modifications compare with the usual approach?

- I have several questions and comments on the following point "In the absence of fresh snow, snow albedo decreased exponentially from its fresh value" (L. 74-75):

  ○ According to Table 1, fresh snow has a fixed albedo. This is a rather crude approximation. Does it make sense? Why not instead using an increase such as the linear increase with snowfall intensity implemented by Boone and Etchevers (2001)? Could the authors reflect on that?
  Reference: Boone, A. and Etchevers, P.: An intercomparison of three snow schemes of varying complexity coupled to the same land-surface model: Local scale evaluation at an Alpine site, J. Hydrometeorol., 2, 374–394, 2001

  ○ Also, fresh snow occurs during winter when no MODIS data are available, how does $B_{aged}$ can be calibrated?

  ○ $A_{aged}$ is the albedo of pure ice, please indicate it in Table 1.

  ○ The snow albedo modelling implement an exponential decrease law that is close to the approach of Douville et al. (1995) but involves, contrary to the aforementioned paper, soil temperature. Where does this formulation come from? Has it been previously published and validated before? Has it been used for ice sheet before as well? I think the modelling of albedo deserves more explanation or justification in the paper.
  Reference: Douville, H., Royer, J., and Mahfouf, J.: A new snow parametrization for the Meteo-France climate model, Clim. Dynam., 12, 21–35, 1995.

- L. 180-182 *"All the simulations performed in this study include two years of model spin-up to allow the snow to accumulate … ensuring correct initial states"*: I assume that the two years of model spin-up are 1998 and 1999 since the period of study is 2000-2017. Am I correct? If so, please indicate it in the manuscript. Also what was the snow depth before model spin-up. Was it 0? Did you allow only snow to accumulate or was melting also occurring? Please provide more details. One key point of a scientific paper is that readers can be able to reproduce the described experiment themselves. The authors should focus on that very point. It would have avoided many questions I have listed.

About the data assimilation approach:

- The approach of selecting three random years (2000, 2010 and 2012) for the optimisation raises the question of the robustness of the approach. While it shows consistent improvement over the whole period of study 2000 – 2017, I wonder if selecting other years would have led to different results especially if, for example, 2000, 2001 and 2002 would have been selected. Could the authors reflect on the robustness of their approach?

- L. 120-122 *"We define the observation error (variance) as the mean-square difference between … but also the model errors"* I am partly unsettled by the authors' stance on observation errors. It breaks the underlying assumption of independence between background, model errors and observation errors of the Bayesian statistical formalism. Nevertheless, I can see why the authors adopt such approach but it deserves more justification. Basically, why using this approach instead of a fixed error variance for MODIS albedo data? Also, it induces more uncertainties at the GriS's edge than in its centre that probably comes from the modelling approach. Does this assumption really make sense?

- About $\tau_{max}$, it was shown by the Morris sensitivity analysis that it has almost no impact. Figure 6b focus on correlations. What about variances? The cost function's Hessian at

minimum can provide local sensitivity analysis (see e.g. . Also its inverse is the analysis covariance matrix. It would be of interest to study analysed variances compared to background variances and evaluate their reduction. Authors show strong anti-correlation between $\delta_c$ and $\tau_{max}$, but associated comments should be weighted with the role of variances.

- Two minimisation algorithms are considered in this study: the gradient-based L-BFGS-B approach and the genetic algorithm. There is no added value of evaluating those two algorithms compared to what was already written in Bastrikov et al. (2018). I think this subject is very much on the side of the authors' work. They should focus instead to the heart of their study: albedo assimilation and remove anything related to the gradient-based L-BFGS-B algorithm including appendix A1.

About the validation:

- The authors acknowledge that as MAR being a model this part of the work cannot be considered as validation thus limiting their study to an evolved proof of concept although still better than a twin experiment. Also the comparison is not independent since MAR atmospheric variables are used as inputs of ORCHIDEE. The study therefore shows the difference of modelling approached in MAR and ORCHIDEE (in its various configurations). What is the objective of such comparison in regards with the objectives of the papers? Again, such question occurs because the context of the study is not stated loudly by the authors.

- I do realize it would be hard work to use independent data to validate the authors' approach but the absence of proper validation really weakens the validity of the approach and therefore the paper in its current version. One possibility could be to use data from GRACE about Greenland mass loss as a way to validate your approach by assuming all the mass loss is carried out by melting (in direct link with SMB). While this assumption is probably excessive as it ignores the impact of outlet glaciers, the comparison with independent data coming from GRACE could solve partially my aforementioned reservations about the validation aspects of this paper.

- Another possibility would be to use in situ data from the PROMICE network as mentioned in the conclusion L. 379-380 "One solution would be to run … lead to issues of scale and representativity". In land surface modelling, it is common for example to compare modelled soil moisture at e.g. 0.25° with in situ measurements, see e.g. Kumar et al. (2019) or Albergel et al. (2020). While there are issues of scale, those comparisons are still by far important and very useful. I do not understand why it could not be done in the authors' context.

References:

- Albergel, C., Zheng, Y., Bonan, B., Dutra, E., Rodríguez-Fernández, N., Munier, S., Draper, C., de Rosnay, P., Muñoz-Sabater, J., Balsamo, G., Fairbairn, D., Meurey, C., and Calvet, J.-C.: Data assimilation for continuous global assessment of severe conditions over terrestrial surfaces, Hydrol. Earth Syst. Sci., 24, 4291–4316, https://doi.org/10.5194/hess-24-4291-2020, 2020.

- Kumar, S. V., Mocko, D. M., Wang, S., Peters-Lidard, C. D., and Borak, J.: Assimilation of remotely sensed Leaf Area Index into the Noah-MP land surface model: Impacts on water and carbon fluxes and states over the Continental U.S., J. Hydrometeorol., https://doi.org/10.1175/JHM-D-18-0237.1, 2019.

- If no independent data are used, then the part of comparing ORCHIDEE and MAR should be strengthen by highlighting the differences in term of modelling for both approaches (with appropriate bibliographical literature) and this comparison should be strongly linked with the Morris sensitivity analysis and with the background context of the study that is definitely missing. Also for SMB and runoff, the difference between MAR and ORCHIDEE clearly occur at the edges of the GrIS (either with standard, tuned or optimised parameters for ORCHIDEE). The parameter optimization for albedo does not make ORCHIDEE closer to MAR (quite the contrary). The previously performed Morris sensitivity analysis could help to understand the mechanisms behind those increased differences and would make a nice discussion section.

**MINOR COMMENTS AND TYPOS**

L. 8 *"This improvement is consistent for all years, even those not used in the calibration step"*. Could the authors rephrase the sentence as I would expect such result otherwise the methodology would not work?

L. 14-16 *"Increased warming … algae growth (Cook et al., 2020)"*. Darkening of GrIS has already observed and expect to worsen and increased impact on GrIS melting. Rephrase accordingly. Also, several missing references, among others:

- Dumont, M., Brun, E., Picard, G., Michou, M., Libois, Q., Petit, J.-R., Geyer, M., Morin, S. and Josse, B.: Contribution of light-absorbing impurities in snow to Greenland's darkening since 2009, Nature Geosci., 7, 509-512, 2014.

- Williamson, C. J., Cook, J., Tedstone, A., Yallop, M., McCutcheon, J., Poniecka, E., Campbell, D., Irvine-Fynn, T., McQuaid, J., Tranter, M., Perkins, R. and Anesio, A.: Algal photophysiology drives darkening and melt of the Greenland Ice Sheet, PNAS, 117(11), 5694-5705, 2020.

- Perini, L., Gostinčar, C., Anesio, A. M., Williamson, C., Tranter, M. and Gunde-Cimerman, N.: Darkening of the Greenland Ice Sheet: Fungal Abundance and Diversity Are Associated With Algal Bloom, Front. Microbiol., 10, https://doi.org/10.3389/fmicb.2019.00557, 2019.

L. 18-19 *"This, in turn, enhances melting, creating feedback to the atmosphere"*. Missing reference to support the statement:

- Le clec'h, S., Charbit, S., Quiquet, A., Fettweis, X., Dumas, C., Kageyama, M., Wyard, C., and Ritz, C.: Assessment of the Greenland ice sheet–atmosphere feedbacks for the next century with a regional atmospheric model coupled to an ice sheet model, The Cryosphere, 13, 373–395, https://doi.org/10.5194/tc-13-373-2019, 2019.

- Box, J. E., Werhlé, A., van As, D., Fausto, R. S., Kjeldsen, K. K., Dachauer, A., Alhstrøm, A. P. and Picard, G.: Greenland Ice Sheet Rainfall, Heat and Albedo Feedback Impacts From the Mid-August 2021 Atmospheric River, Geophys. Res. Lett., https://doi.org/10.1029/2021GL097356, 2022.

L. 22: *"… it is crucial that it is accurately simulated in*  *land surface models (LSMs) …"*

L. 39-40: *"Examples of DA used for parameter estimation … in snow modelling are less common. Bonan et al. (2014) … "*. The reference is not about snow modelling but on ice sheet initialization and DA or inverse modelling is well known in this field, see the following paragraph in Bonan et al. (2014) "MacAyeal (1992) and MacAyeal (1993) introduced control methods to infer basal drag in ice-stream models, using in particular the self-adjoint property of such models, leading to many application papers (Rommelaere and MacAyeal, 1997; Vieli and Payne, 2003), and later for full Stokes models (Morlighem et al., 2010; Jay-Allemand et al., 2011). Later on, many DA and inverse methods were introduced in glaciology. The Best Linear Unbiased Estimation (BLUE) and Optimal Interpolation (OI) methods were introduced by Arthern (2003) and Berliner et al. (2008). The Robin inverse method due to Chaabane and Jaoua (1999) has been introduced by Arthern and Gudmundsson (2010) for ice sheet models, and finally Heimbach and Bugnion (2009) presented the first adjoint ice sheet model derived automatically." Most previous references focuses on estimating basal friction or basal velocities as parameters. Regarding parameter estimation for ice sheet mass balance, you can see:

- Bonan, B., Nodet. M., Ozenda, O. and Ritz, C.: Data assimilation in glaciology, in Advanced Data Assimilation for Geosciences, Lecture Notes of the Les Houches School of Physics: Special Issue June 2012 (Edited by Blayo, E., Bocquet, M., Cosme, E. and Cugliandolo, L. M.), 577-584, Oxford University Press, 2014.

L. 48-49: *"However, with large amounts of data, … the multisite approach is common"*: Debatable statement. One of the main reason of the multisite approach is commonly used is that the set of optimal parameters or parametrizations for various sites in LSMs can differ significantly due to soil properties (soil texture, water potential, hydraulic conductivity …) and land cover (vegetation variables). Please soften the previous sentence accordingly.

L. 64: authors indicate that CMIP 6 version of ORCHIDEE is used in this paper: add reference publication(s) for this version in addition to the historic paper of Krinner et al. (2005). Also authors mention in the code availability section that ORCHIDEE vAR6 is employed. Could the authors harmonize notations between both paragraphs?

L. 73-74 *"we computed the mean of albedo in both visible (VIS) and near-infrared (NIR) spectral domains"*. Please indicate that this is to be in accordance with MODIS data. Also the description of albedo following this sentence does not distinguish VIS and NIR spectral domains. I do not think spectral domains intervene in the computation of modelled albedo. Rephrase sentence L. 73-74 in order to state that your model does not distinguish VIS and NIR albedo.

Section 2.2: Could the authors cite the paper(s) associated with MAR Version 3.11.4? Gallée and Schayes (1994) is rather outdated for this version.

Section 2.3: Please mention that this dataset do not include data from the Aqua satellite as explained in Box et al. (2017). People familiar with MODIS datasets tend to expect data coming from both Terra and Aqua satellites.

L. 106-108 *"Finally, in this dataset, … in the April values"*. This statement is rather confusing, please rephrase.

L. 114-120 *"Bayesian statistical formalism (Tarantola, 2005)"*. The formulation of the cost function can also be seen as an optimal control problem without any assumption on probabilistic distributions. This has given the basis of 3D and 4D-Var approach, see for example Nichols (2010). Could the authors rephrase L.114 to L.120 to play down the emphasis on the Bayesian statistical formalism?

- Reference: Nichols, N. K.: Mathematical concepts of data assimilation, in: Data assimilation: making sense of observations, edited by: Lahoz, W., Khattatov, B. and Menard, R., Springer-Verlag, Berlin, Germany, 13–40, 2010

L. 163 *"they correspond to ablation areas"* Most parts of Greenland nowadays experience ablation during summer. Edges are where strong ablation occurs. Please rephrase accordingly.

L. 168 *"… into  ORCHIDEE, where … "*

L. 174-176 *"They were also the pixels with the largest errors when compared … with RMSD greater than 0.1"*. When this calculation is performed? Before calibration or after? Can the author explain where does this number comes from? It can be simply done by referring to a subsequent section of the paper.

L. 181 *"… to allow  snow to accumulate …"*

L. 209 *"Bayesian framework"* see previous comment on the Bayesian term

About Figure 2: *"currently operational ORCHIDEE version"* By currently operational ORCHIDEE version, did the author mean ORCHIDEE with parameters set at default values (as in Table B1)? To my knowledge, the way albedo is modelled and the new "icy" type cannot be called "operational" yet. Could the authors modify the legend of Figure2 to reflect this point?

L. 223 *"… affect by  albedo parameters …"*

L. 297 *"Bayesian framework"* see previous comment on the Bayesian term

L. 302 I think the text should refer to Figure 4a instead of Figure 6a here.

L. 308 Replace *"omega"* by ω and *"beta"* by β.

L. 320 *"… different parameter setS on modelling …"*

L. 326-327 *"Compared to MAR, the manually … ORCHIDEE performs best at simulating SMB"*. Best performance does not really make sense in the context of comparing two models. This sentence and the rest of the section should be rewritten keeping this fact in mind.

L. 347 *"Bayesian optimisation"* see previous comment on the Bayesian term

L. 347-348 *"However, we overfitted to albedo with no other data"*. This statement raises the question of prescribed observation error variances. Would other prescribed values have made the impact of optimized parameters for ORCHIDEE more in line with MAR? Or is this question related to modelling differences? Could the authors reflect on that question in the paragraph?

L 360-361 *"When we cannot further improve … this can point to structural deficiencies in the model"*. I tend to disagree with this statement. Parameter estimations can sometimes hide structural model deficiencies, i.e. you may obtain the right results but for the wrong reason. Could the authors weight on that comment?

L. 361-362 *"For example, we cannot capture the different albedos in the north and the south of the ice sheet with the current processes represented"* This problem might also come from that the

author assume the same set of parameters for the whole Greenland. A multisite approach may have reduce this problem (perhaps for the wrong reasons). Could the authors reflect on that question?

L. 375 "*There is an urgent need for data producers to provide this uncertainty, ideally at each time step*". I could not agree more. The authors can mention this following reference to strengthen their statement:
- Merchant, C. J., Paul, F., Popp, T., Ablain, M., Bontemps, S., Defourny, P., Hollmann, R., Lavergne, T., Laeng, A., de Leeuw, G., Mittaz, J., Poulsen, C., Povey, A. C., Reuter, M., Sathyendranath, S., Sandven, S., Sofieva, V. F., and Wagner, W.: Uncertainty information in climate data records from Earth observation, Earth Syst. Sci. Data, 9, 511–527, https://doi.org/10.5194/essd-9-511-2017, 2017.

L. 379-380 "*One solution would be to run … lead to issues of scale and * REPRESENTATIVENESS".

L. 395 Rephrase the sentence to replace *"better"* by "more consistent with MAR outputs"

L. 398-404 I have some reservations with the statements written in the paragraph. Would the idea behind using all these satellite datasets be to replace the modelling of the Greenland ice sheet ice dynamics? Could the authors temper those statements?

---

## Author Response (AR2)

**Reviewer 1**

The work in this paper is motivated by the critical nature of climate change and its effect on the Greenland ice sheet. It is an interesting study presenting some very detailed work on improving a particular (ORCHIDEE) land surface model with the known techniques of parameter optimisation (for that model) by use of external, independent observations. These observations are the retrieved snow albedo product from MODIS. A regional model simulation produced output that was also used to compare with the ORCHIDEE model run with the different parameter sets. I think that the paper should be published, subject to some minor revisions to clarify the work for the readers.

Please see the attached pdf which comments on particular lines to aid the authors in making both small and larger changes. I have made some suggestions on where definitions and citations would be useful among other recommendations and critical comments. I hope that the authors find this useful.
We would like to thank the reviewer for taking the time to read and comment on our manuscript.

I made a comment about the accessibility of colour in one one figure (jet is not a good choice for communicating information). The following sites contain useful information, tips and tools on appropriate use of colour in illustrations:
https://www.nature.com/articles/s41467-020-19160-7
https://www.ascb.org/science-news/how-to-make-scientific-figures-accessible-to-readers-with-color-blindness
https://towardsdatascience.com/two-simple-steps-to-create-colorblind-friendly-data-visualizations-2ed781a167ec
In addition, here is very convincing talk about why the Python standard map is now "viridis" and why it is better than jet.
https://www.youtube.com/watch?v=xAoljeRJ3lU
Thank you for highlighting this issue, we have changed the colour of the plot to viridis as suggested.

In summary, my recommendation is for minor revisions to address issues of clarity in the communication of the work.
We thank the reviewer for this recommendation. We have implemented the suggestions from the PDF, which we agree improves the readability and clarity of the work.

**Reviewer 2**

OVERVIEW

This paper aims to improve simulating snow albedo by the ORCHIDEE land surface model via assimilation of MODIS snow albedo retrievals and parameters calibration through ORCHIDAS. The domain of study is the Greenland Ice Sheet (GrIS). ORCHIDEE is used in offline mode forced using atmospheric data coming from the MAR model. The optimisation is performed over three random years (2000, 2010 and 2012) taken over the period of study 2000-2017 while the improvement of snow albedo is checked over the whole period of study. An increased weight is given to pixels defining the edges of the GrIS compared to other pixels in order to improve albedo simulations over those areas crucial in term of mass loss. Moreover, a comparison between outputs of ORCHIDEE and MAR on surface mass balance, runoff and sublimation is carried out as well as a Morris sensitivity analysis of various parameters related to snow albedo and snow density relative to several outputs such as surface temperature, sensible and latent heat fluxes in addition to the previously mentioned outputs

We would like to thank the reviewer for taking the time to read the manuscript and comment on it, as well as for their faith in our ability to improve the paper for publication. We believe the different clarifications asked for and addition of an in situ evaluation will significantly strengthen the manuscript.

GENERAL COMMENTS

The paper covers the timely question of snow/ice albedo modelling over the GrIS in the context of increased melt in links with global warming. Here the chosen approach is to assimilate satellite derived albedo retrievals, which only few studies have performed before. As such, the novelty aspects of the paper are self evident. Therefore, such work would clearly deserve to be published. Nevertheless, I have some reservations about the methodology and I especially find the paper difficult to read. As indicated by one previous referee, "The article reads like a description of the research in the way it was conducted.". Unfortunately, after the first revision, this problem still remains. My main issues with the paper are the following:

- The goal of the paper is straightforward: improve snow albedo by assimilation. But why do we want to do that? In which context? Global land surface modelling, if so, why using MAR instead of ERA5 as atmospheric forcing? Or are the targets are ice sheets and large glaciers? During all the several readings I made, I asked myself what were the purposes of such study and I could not find the answer anywhere in the introduction where normally one would expect to find such information.
  We acknowledge that we need to be more explicit in describing the context and motivations of our study focused on modelling the Greenland ice sheet, as well as our choice of the MAR forcing. Additions to the manuscript are described below in response to the relevant comments.

- The methodology clearly misses validation using independent datasets. Modelled albedos are compared with MODIS data that are partially assimilated (3 years over 18 years). Then ORCHIDEE outputs of surface mass balance, runoff and sublimation are compared with outputs from MAR, while ORCHIDEE is forced using atmospheric data coming from MAR itself. Obviously both model outputs are related, the main difference would come from the modelling approach of snow, ice and albedo in ORCHIDEE and MAR. As such, MAR cannot be considered as a reference as stated by the authors. I do, however, acknowledge that the authors raise some reservations about the limitation of the comparison in the paper.

  We acknowledge that evaluation against independent data is lacking. Therefore, we have added a section evaluation against in situ PROMICE data. This is further discussed in the validation section of the comments below.

- Several peripheral considerations (such as the test of the two minimisation approach, LBFGS-B and genetic algorithm) degrade the readability of the paper while providing very little novelty in terms of science.

  All mentions of L-BFGS-B have been removed from the manuscript for added clarity.

As I indicated beforehand, this paper cannot be published in The Cryosphere in its current version. Nevertheless, I do think the authors, who are renowned experts in their fields of expertise, have the ability to improve the paper to reach the publication stage. It would be unfortunate that such good science would, in the end, not be published. I list my comments and questions section by section below:

SPECIFIC COMMENTS

About the objectives of the paper:

- Please specify loudly the context and the purposes of the study in the introduction. Is the goal of the study to focus on ice sheets and large glaciers? If so, why? If not, do the authors focus on global or large scale climate simulations? If so, again why? Please provide also adequate references to justify your choices.

  Modelling the Greenland Ice Sheet is vital in understanding sea-level rise, amongst other factors impacted by a changing climate. Both dynamical effects and surface processes drive Greenland's evolution. However, recent studies show that surface mass balance changes dominate mass balance changes (The IMBIE team, 2020; Ryan et al., 2019; van den Broeke et al., 2016). To correctly represent the surface mass balance and its components (sublimation and runoff), it is important to simulate the physical processes within the snowpack. These depend on the surface energy balance and, therefore, on the albedo. The snow albedo model in ORCHIDEE has only ever been calibrated for vegetation and bare soil - never for glaciers or ice sheets.

  This work is part of ongoing work at LSCE to integrate an ice sheet model into ORCHIDEE. Currently, when used as part of the larger IPSL Earth System Model, the

atmospheric component (LMDZ) treats the ice sheets through a rudimentary one-layer scheme and a fixed albedo. There is a desire for ORCHIDEE to model all types of continental surfaces. The modelling of the snow surface processes as it is done in ORCHIDEE would be a more suitable, coherent and explicit replacement to the original LMDZ scheme. This work will be documented fully in Charbit et al. (prep). Optimising parameters of the albedo of this snow model is a vital step in this development, given the importance of albedo, and this Raoult et al. paper discusses the techniques we can use to perform this parameter optimisation. Therefore this paper acts as a step in the development of the ORCHIDEE snow model with an application over the Greenland ice sheet. As discussed in the conclusions, ORCHIDAS is well adapted to the optimisation purpose, but, given the complexity of modelling the surface mass balance, more structural changes are needed to ORCHIDEE first, and further optimisations should consider more data in addition to the satellite albedo.

The second paragraph of the introduction has been expanded as follows:

*Both dynamical effects and surface processes drive Greenland's evolution. However, recent studies show that surface mass balance changes dominate mass balance changes (The IMBIE team, 2020; Ryan et al., 2019; van den Broeke et al., 2016). To correctly represent the surface mass balance and its components (sublimation and runoff), it is important to simulate the physical processes within the snowpack. These depend on the surface energy balance and, therefore, on the albedo.* Given the importance of *this* albedo, it is crucial that it is accurately simulated in land surface models.

In addition, the final paragraph of the introduction has been expanded as follows:

"Using MODIS snow albedo, in this study, we use DA for parameter estimation to improve the albedo parameterisation *over ice sheets* inside the ORCHIDEE LSM (Krinner et al., 2005). *While albedo parameters in ORCHIDEE have been optimised for vegetation and bare soil, this will be the first study optimising them for ice sheets. Our target area from this study is the Greenland Ice Sheet. This study is the first test of applying the ORCHIDEE data assimilation system over ice sheets to improve modelling albedo and, in turn, the surface mass balance of the ice sheet.* Instead…"

Reference:

Van den Broeke, Michiel R., et al. "On the recent contribution of the Greenland ice sheet to sea level change." *The Cryosphere* 10.5 (2016): 1933-1946.

The IMBIE team, "Mass balance of the Greenland Ice Sheet from 1992 to 2018." *Nature* 579, no. 7798 (2020): 233-239.

Ryan, J. C., et al. "Greenland Ice Sheet surface melt amplified by snowline migration and bare ice exposure." *Science Advances* 5.3 (2019): eaav3738.

- Justify more adequately the use of MAR in relations to your goals either in the introduction and section 2.2 describing MAR. MAR is indeed a fine atmospheric model well adapted to Greenland but by using MAR, the authors make their approach less replicable to other parts of the world if their goal is to fine tuning parameters for modelled albedo for ice sheets and large glaciers as optimal parameters found are likely to depend on atmospheric inputs.

  It is true that MAR is less replicable in other parts of the world. However, as now explicitly stated in the introduction as per the previous comment, our target region is Greenland. For this region, MAR has been shown to outperform reanalysis products such as ERA5 (Delhasse et al., 2020). We have added the following sentence to the end of section 2.2. describing MAR:

  > "*MAR was specifically developed for polar regions and offers good performances for the calculation of SMB and its components. Furthermore, it has been shown to outperform reanalysis products such as ERA5 (Delhasse et al., 2020), especially in providing the near-surface temperature in summer which play a critical role in representing snow and ice processes.*"

  Reference:
  > Delhasse, A., Kittel, C., Amory, C., Hofer, S., van As, D., S. Fausto, R., and Fettweis, X.: Brief communication: Evaluation of the near-surface climate in ERA5 over the Greenland Ice Sheet, The Cryosphere, 14, 957–965, https://doi.org/10.5194/tc-14-957-2020, 2020.

- The authors starts to talk about runoff, sublimation and surface mass balance (SMB) from L. 92-93. This should be specify in the introduction why those quantities are considered in the discussion section of the paper.

  The ultimate goal of the ORCHIDEE snow model developments is to have the best possible representation of the snow energy budget and SMB (and thus of its components). While accumulation depends directly on forcing, the processes related to compaction and ablation are the ones modelled. Hence our interest is the effect of the optimisation on the modelling of the runoff, sublimation and SMB. We have added the following to the introduction:

  > "*To correctly represent the surface mass balance and its components (sublimation and runoff)...*"
  > "*This study is the first test of applying the ORCHIDEE data assimilation system over ice sheets to improve modelling albedo and, in turn, the surface mass balance of the ice sheet.*"

  And to section 3.2:
  > "*We are especially interested in the impact of the albedo parameters of the surface mass balance and its components (sublimation and runoff)*"

- Also MAR includes a modelled albedo. How does modelling in ORCHIDEE compare with MAR's? Might be worth comparing both if possible

MAR albedo has been added to Fig 2, and the text expanded to discuss how MAR albedo compares to MODIS and ORCHIDEE.

- Mass loss in Greenland is not only due to snow melting but also ice dynamics in outlet glaciers that are located in many edge pixels. This should be mention somewhere as it is a limitation of the authors' work (no ice dynamics considered) in the introduction with adequate references such as:
    - Aschwanden, A., Fahnestock, M. & Truffer, M. Complex Greenland outlet glacier flow captured. Nat Commun 7, 10524 (2016). https://doi.org/10.1038/ncomms10524
    - Khan, S.A., Bjørk, A.A., Bamber, J.L. et al. Centennial response of Greenland's three largest outlet glaciers. Nat Commun 11, 5718 (2020). https://doi.org/10.1038/s41467- 020-19580-5

The snow model optimised in this study has been developed to simulate mass loss from surface processes only, not to simulate the total mass loss. It is not a dynamic ice sheet model. This has been made clearer in the introduction when stating the context and purpose of the study.

- Darkening of the GrIS is mentioned in the introduction as a very important phenomenon for albedo. Nevertheless, this darkening is not explored anywhere in the paper. The ability of the authors' approach to capture this darkening could be studied in the result section.

This model does not explicitly take into account the deposition of aerosols, algae and dust. In addition, although snow ageing, linked to the metamorphism of the snow, is taken into account, metamorphism itself is not explicitly modelled. These limitations have been added to the conclusions in paragraph discussing structural changes (starting L359):

> *"Processes linked to the darkening on the ice sheet (e.g., deposition of aerosols, algae and dust) also need to be considered in future developments of the model."*

Indeed, work on the deposition of dust and its impact on snow albedo is under development.

- Since this paper is about albedo assimilation, a longer paragraph dedicated to that very subject would be much appreciated. Also several references are missing, among others:
    - Dumont, M., Durand, Y., Arnaud, Y. and Six, D.: Variational assimilation of albedo in a snowpack model and reconstruction of the spatial mass-balance distribution of an alpine glacier, J. Glaciol., 58(207), 151-164, doi: 10.3189/2012JoG11J163, 2014.
    - Cedelnik, J., Carrer, D., Mahfouf, J.-F. and Roujean, J.-L.: Impact Assessment of Daily Satellite-Derived Surface Albedo in a Limited-Area NWP Model, J. Appl. Meteorol. Clim., 51(10), 1835-1854, https://doi.org/10.1175/JAMC-D-11-0163., 2012.
    - Boussetta, S., Balsamo, G., Dutra, E., Beljaars, A. and Albergel, C.: Assimilation of surface albedo and vegetation states from satellite observations and their

impact on numerical weather prediction, Remote Sens. Environ., 163, 111-126, https://doi.org/10.1016/j.rse.2015.03.009, 2015.

○ Geppert, G.: Analysis and application of the ensemble Kalman filter for the estimation of bounded quantities, PhD thesis, Universität Hamburg, Hamburg. Doi: 10.17617/2.2161673, 2015.

The albedo DA paragraph in the introduction has been expanded as follows:

"Several studies have used remotely sensed albedo for DA in LSMs. *Due to albedo's influence on the partitioning of the surface energy fluxes and the subsequent effect on the development of planetary boundary conditions and clouds (Pielke & Avisser, 1990), some studies have focused on the impact assimilating surface albedo has on numerical weather prediction (e.g., Cedelnike et al., 2014; Bousseta et al., 2015). Others have mainly used remotely sensed data to derive new vegetation and soil background albedo parameters to use in land surface models (e.g., Liang et al., 2005; Houldcroft et al., 2009). There are also a number of examples of using snow albedo to improve snow models.* Malik et al. (2012) used MODIS-based snow albedo and direct insertion methodology in the Noah LSM over three sites in Colorado to improve simulated snow depth and snow season duration. Satellite-based albedo data was also used by Wang et al. (2015) to calibrate the ORCHIDEE (ORganizing Carbon and Hydrology in Dynamic Ecosystems, Krinner et al., 2005) LSM and investigate the impacts of albedo assimilation on offline and coupled model simulations. *Dumont et al., (2014) assimilated remotely sensed albedo in the Corcus snowpack model (Vionnet et a., 2012) to improve the modelling of the spatial distribution of the glacier mass balance.* Navari et al. (2018) *further* improved the Crocus model using satellite-derived albedo to improve surface mass balance (SMB) along Greenland's Kangerlussuaq transect."

Additional references:

Liang, X.-Z., et al. (2005), Development of land surface albedo parameterization based on Moderate Resolution Imaging Spectroradiometer (MODIS) data, *J. Geophys. Res.*, 110, D11107, doi:10.1029/2004JD005579.

Houldcroft, Caroline J., et al. "New vegetation albedo parameters and global fields of soil background albedo derived from MODIS for use in a climate model." *Journal of Hydrometeorology* 10.1 (2009): 183-198.

- A plan at the end of the introduction describing the sections developed in the paper would improve the readability of the paper to a great extent.

The following has been added:

"The paper is organised as follows. Methods and data, including the details about the ORCHIDEE land surface model and its data assimilation framework, driving and observational datasets, and performed experiments, can be found in Sect. 2. Section 3 lists the results, starting with an assessment of the prior and a sensitivity analysis of the main parameters. These are followed by results of the

*main experiments, and an evaluation over PROMICE in situ sites. In Sect. 4, we
look at the impact of the optimisation of the modelling of the SMB of the GrIS, as
well as the different SMB components. Finally, the discussion and conclusions
can be found in the Sect. 5."*

About the methodology:

- The set of parameters is optimized over the whole GrIS but albedo conditions varies greatly between the interior and the edges of the GrIS as stated in the paper. Does it make sense to follow this approach instead of a multisite approach? Also, Figure 4b shows differences between Northern and Southern parts of the GrIS.
  We did one optimisation over the whole of the GrIS because the aim was to have one set of parameters that works across the ice caps and with different climates. An optimisation on selected sites might have given better results, but that could have meant optimising the specificities of those sites and not having a parameter set robust under different climatic conditions. Furthermore, selecting representative sites, the number of sites to use and the number of parameter sets to have in the end comes with its challenges. There would be a lot of additional considerations and choices to be made, which may lead to different outcomes. In ORCHIDEE, one parameter set is currently used and we are able to greatly improve the model albedo by just optimising that one set.

- The Morris sensitivity analysis does not intervene in the choice of optimized parameters. It should be instead include in a discussion section after the analysis of results. Also it involves parameters that are not optimized. The Morris sensitivity analysis and the comparison between MAR outputs and ORCHIDEE outputs could be merged in a same discussion section as they tend to complete each other. This discussion should also be pointed out in the introduction of the paper. Also, the list of parameters of interest should have be provided in the ORCHIDEE land surface subsection. Since the authors focus not only on albedo per se but also on rate of density change and parameters controlling surface mass balance and runoff, authors should provide more information on the snow model employed in ORCHIDEE or at least point towards adequate references. This would help the readers to understand more clearly the authors' objectives
  After the first round of reviews, we moved the sensitivity analysis to the forefront of the manuscript, as requested by one of the reviewers. We believe that there are arguments to be made in both cases. This type of analysis makes sense at the beginning of the manuscript since it is often the first step in data assimilation. However, since the Morris sensitivity analysis does not intervene in the choice of optimised parameters and the fact that we introduce parameters from additional parameterisations not discussed elsewhere in the manuscript, it makes sense to have it at the end of the manuscript as part of the discussion. After going back and forth on this, we have decided to return the analysis to the discussion since a) it will help alleviate concerns around the choice to keep tau_max, b) allow us to make links between the comparison ORCHIDEE-MAR outputs and parameter sensitivities and posterior values, and c) avoid confusion between the

parameters of the albedo parameterisations and the additional ones considered. Further to this, we believe the additional parameters used in the sensitivity analysis are best kept in the appendix to avoid confusion with the parameters used in the optimisation. During the first round of reviews, we had not yet included the comparison MAR-ORC and so the sensitivity analysis seemed out of place. Now with this comparison, the sensitivity analysis can be used to help explain some of the behaviours observed. The following as been added to the SA section:

> When comparing different ORCHIDEE runs to MAR (Sect. 4.1), we saw that sublimation was the output most impacted by the different parameter sets. This was especially notable at the centre of the ice sheet. This sensitivity analysis highlights that sublimation at the centre of the ice sheet is most sensitive to the Bdec and τdec parameters, which are changed in the optimisation to lower the decay term therefore increase albedo. In contrast, for runoff and SMB, both of which show no spatial variability over the middle of ice sheet in Fig. 7, the v2 parameter from the viscosity parameterisation is more important. However, this parameter was not optimised in this study. Nor were other parameters from the viscosity parameterizations, to which sublimation, runoff and SMB are sensitive, especially at the edges. Although we do get some variation in runoff and SMB in the different ORCHIDEE runs (Sect. 4.1), since these are concentrated at the edges, it is possible that by optimising these viscosity parameters we would better fit MAR outputs.

The added plan in the introduction highlights the discussion between the MAR and ORCHIDEE outputs and the concluding sensitivity analysis. For a detailed response about concerns linked to the snow model and adequate references, see below in response to the snow and albedo modelling section of the reviews.

- $\tau_{max}$ seems to have almost no influence for albedo according to the Morris sensitivity analysis, why keeping it for optimization?
  While the Morris is a useful tool to get an idea of sensitivities, it remains exploratory with some caveats (dependant on ranges used and the number of iterations, its treatment correlated parameters etc). Since we are working with a small number of parameters, we decided to include all from the albedo parameterisation since these are all varied in the manual tuning experiments.

About MODIS data:

- About MODIS albedos, it is unclear to me if there are any reliable observations during winter time. Authors exclude data from November to February in their data assimilation system but later states albedo is improved in winter times in the result sections. This is rather confusing. I suggest you exclude all albedo comparison for winter period as I do not think they make sense (see e.g. sentence L. 280-281 "We also see that the optimisations improve the fit … in the optimisation"). Alternatively, authors can explicit in the manuscript their methodology regarding albedo during winter time.
  Yes, we agree - we forgot to remove this sentence. This has now been removed so that no comparison to winter is considered in the manuscript.

- How do the authors aggregate MODIS data from the original grid to the MAR grid (just an average or something more sophisticated)? That should be made more explicit in the text.

  The MODIS data was aggregated using bilinear interpolation. L116 has been expanded to state this:

  "...further aggregating these data *using bilinear interpolation* to the resolution of the ORCHIDEE outputs, imposed by the meteorological forcing files (20 km)."

About the snow and albedo modelling and spin up:

- L. 69-71: "For the new icy soil type, … those of the loam soil type because it is the dominant soil type in the non-ice-free reguibs around the GrIS (Fischer et al., 2008)". Does it make sense to assume that basically, the icy soil type is impermeable (porosity and saturation are equal) for the edge of ice sheets?

  Ice is an almost entirely impervious medium, whether in the centre of the cap or at the edges. However, when crevasses are formed at the surface due to surface meltwater or liquid precipitation, surface water may penetrate at depth and flow through channel networks until reemerging at the glacier front. We acknowledge that this process (called a "moulin" in glaciology) is not accounted for in ORCHIDEE as the model does not represent the lateral water transport - snowmelt and icemelt are added to runoff which can either infiltrate (on flat terrains and if a part of the grid is composed of non-glaciers areas) or be transferred to the river network. We have added the following to L71:

  "This amounts to considering that ice is an impermeable medium. However, it does not allow the representation of processes such as moulins where water seeps through a network of galleries because the model does not simulate the lateral transport of water."

- What is it classically used for ORCHIDEE in CMIP experiments and how do your modifications compare with the usual approach?

  Until now, ORCHIDEE did not consider ice-covered surfaces (including in CMIP simulations). In fact, when coupled with the atmospheric LMDZ model, it is LMDZ with a rudimentary one-layer scheme and a fixed albedo that handles the ice caps.

- I have several questions and comments on the following point "In the absence of fresh snow, snow albedo decreased exponentially from its fresh value" (L. 74-75):
  - According to Table 1, fresh snow has a fixed albedo. This is a rather crude approximation. Does it make sense? Why not instead using an increase such as the linear increase with snowfall intensity implemented by Boone and Etchevers (2001)? Could the authors reflect on that?
    Reference: Boone, A. and Etchevers, P.: An intercomparison of three snow schemes of varying complexity coupled to the same land-surface model: Local scale evaluation at an Alpine site, J. Hydrometeorol., 2, 374–394, 2001

  Two snow albedo schemes were implemented in ORCHIDEE and tested in Wang et al., (2012); the first is the formulation from Boone and Etchevers (2001) and the second the

formulation from Chalita and Le Treut (1994) which was used as a standard in the model. This latter formulation was found to work better when evaluated against in situ sites. Therefore, the Chalita and Le Treut (1994) albedo scheme is the one used in ORCHIDEE. This citation has been added to the text.

In our model, the albedo of fresh snow is constant but is different over GIS compared to the land surfaces. The albedo value of old snow differs widely from one location to the other as a function of the environmental conditions and the state of the snowpack. In a very near future, we intend to use satellite observations to evaluate this value, and see if parameterizations such as the one proposed by Boone and Etchevers (2001) could be valuable.

Reference:
Chalita, S., and H. Le Treut. "The albedo of temperate and boreal forest and the Northern Hemisphere climate: a sensitivity experiment using the LMD GCM." *Climate Dynamics* 10 (1994): 231-240.

- Also, fresh snow occurs during winter when no MODIS data are available, how does $B_{aged}$ can be calibrated?
  Since $B_{aged}$ is constant in ORCHIDEE, we only have one value for the whole year.

- $A_{aged}$ is the albedo of pure ice, please indicate it in Table 1.
  $A_{aged}$ is the albedo of old snow, this has been added to Table 1.

- The snow albedo modelling implement an exponential decrease law that is close to the approach of Douville et al. (1995) but involves, contrary to the aforementioned paper, soil temperature. Where does this formulation come from? Has it been previously published and validated before? Has it been used for ice sheet before as well? I think the modelling of albedo deserves more explanation or justification in the paper. Reference: Douville, H., Royer, J., and Mahfouf, J.: A new snow parametrization for the Meteo-France climate model, Clim. Dynam., 12, 21–35, 1995.
  As described in the previous answer, this formulation comes from Chalita and Le Treut (1994) and tested in Wang et al., (2013). This citation has been added to the text. The exponential decrease depends on snow age which is itself dependent on the function fage. The expression of fage was initially implemented in the LMDZ atmospheric general circulation model to take into account the slowing down of metamorphism (and thus of the snow aging) in the polar regions due to the extremely cold temperatures. This formulation has been tested at Dome C (East Antarctica) within the framework of a PhD thesis, but to our knowledge, it has never been published in a peer-reviewed journal. Subsequently, it has been implemented in the ORCHIDEE snow scheme for glaciated areas. In Douville et al. (1995), a weak decrease is also taken into account during cold days, which is somehow equivalent to our formulation, except that "cold days" are explicitly recognized thanks to Tsoil. We would like to remind here, that, so far, the snow

model has never been applied to ice sheets or glaciers. As a consequence, there is no reference related to this formulation.

- L. 180-182 "All the simulations performed in this study include two years of model spin-up to allow the snow to accumulate … ensuring correct initial states": I assume that the two years of model spin-up are 1998 and 1999 since the period of study is 2000-2017. Am I correct? If so, please indicate it in the manuscript. Also what was the snow depth before model spin-up. Was it 0? Did you allow only snow to accumulate or was melting also occurring? Please provide more details. One key point of a scientific paper is that readers can be able to reproduce the described experiment themselves. The authors should focus on that very point. It would have avoided many questions I have listed.
The two years of spin up were the two years preceding each year simulated; i.e., 1998 and 1999 for 2000, 2008 and 2009 for 2010, and 2010 and 2011 for 2012. The model was run normally, i.e., allowed for accumulating and melting, from a snow depth of 0. The precision has been added to the text:
  > "*In each case, the two years preceding the years of study were used in the spinup and the model normally over these years (i.e., allowing for accumulation and melting) from an initial snow depth of 0.*"

About the data assimilation approach:

- The approach of selecting three random years (2000, 2010 and 2012) for the optimisation raises the question of the robustness of the approach. While it shows consistent improvement over the whole period of study 2000 – 2017, I wonder if selecting other years would have led to different results especially if, for example, 2000, 2001 and 2002 would have been selected. Could the authors reflect on the robustness of their approach?
It is true that this is a limitation of the study. To best answer, we would have to repeat the experiment with a different subset of years - however, redoing the simulation would require a lot of computational time. Instead, we have added the following to the discussion:
  > "*For the optimisations, we also selected three random years instead of the full time series. However, it is possible that a different subset of years would give different results. Nevertheless, given the consistent improvement found over the whole period, we do not think the results would be too different.*"

- L. 120-122 "We define the observation error (variance) as the mean-square difference between … but also the model errors" I am partly unsettled by the authors' stance on observation errors. It breaks the underlying assumption of independence between background, model errors and observation errors of the Bayesian statistical formalism. Nevertheless, I can see why the authors adopt such approach but it deserves more justification. Basically, why using this approach instead of a fixed error variance for MODIS albedo data? Also, it induces more uncertainties at the GriS's edge than in its

centre that probably comes from the modelling approach. Does this assumption really make sense?

We acknowledge that our method for defining the observation errors is not ideal. It is true that we want the observations errors in **R** to be independent from the parametric error in **B** and by using a prior simulation to describe **R,** this is not assured. However, **R** must contain both the observation error and the model structural error - this estimation of structure error is crucial. If we only used the error variance of MODIS, we would be neglecting this critical structural error. In an ideal world, we would be able to estimate the structural error properly, for example, by comparing a large number of different models. Since we cannot do this, we use the prior model as a proxy.  Furthermore, the error in the parameters is small compared to the structural error. Since the edges are where we have the most uncertainty - both in terms of modelled processes and retrievals errors - it makes sense for the errors to be larger there.

We have added the following to the text:

> "Although not ideal, this approach is common since it is one of the only ways we can assess the model structural error, which is a large contributor to the **R** matrix.

- About $\tau_{max}$, it was shown by the Morris sensitivity analysis that it has almost no impact. Figure 6b focus on correlations. What about variances? The cost function's Hessian at minimum can provide local sensitivity analysis (see e.g. . Also its inverse is the analysis covariance matrix. It would be of interest to study analysed variances compared to background variances and evaluate their reduction. Authors show strong anti-correlation between $\delta_c$ and $\tau_{max}$, but associated comments should be weighted with the role of variances.

Thank you for this suggestion. We have now calculated the Hessian at the optimum and used this to calculate the reduction in parameter uncertainty. These values match up with the sensitivity analysis in Fig. 3 where Bdec was found to be the most sensitive parameter and tau_max the least. These results impact the correlation discussion, which has now been caveated by these results. The following text has been added, first to the ORCHIDAS section 2.3:

> ### 2.3.4 Posterior uncertainty
>
> Assuming Gaussian prior errors and linearity of the model in the vicinity of the solution, the posterior error covariance matrix of the parameters, **A**, can be approximated by
> $$\mathbf{A} = [\mathbf{M}^T\mathbf{R}^{-1}\mathbf{M}+\mathbf{B}^{-1}]^{-1}$$
> where **M** is the model sensitivity (Jacobian) at the minimum of J(**x**) (Tarantola, 2005).

And second to the Posterior parameter section:

> "However, this relationship is seen to not be critical when we consider the variance at the optimum. We can see that $\tau_{max}$ remains unconstrained by the optimisation. The reduction parameter uncertainty is small - the lowest of all the

parameters. The other parameters show high levels of parameter uncertainty reduction, showing they are highly contained by the optimisation, with $B_{dec}$ reducing the most. This analysis mirrors the results of the sensitivity analysis in Sect. 2.3.2 - the most sensitive parameter shows the largest reduction in uncertainty, and the least sensitive show the smallest reduction."

- Two minimisation algorithms are considered in this study: the gradient-based L-BFGS-B approach and the genetic algorithm. There is no added value of evaluating those two algorithms compared to what was already written in Bastrikov et al. (2018). I think this subject is very much on the side of the authors' work. They should focus instead to the heart of their study: albedo assimilation and remove anything related to the gradient-based LBFGS-B algorithm including appendix A1.
  We have now removed all mentions of L-BFGS-B from the manuscript.

About the validation:

- The authors acknowledge that as MAR being a model this part of the work cannot be considered as validation thus limiting their study to an evolved proof of concept although still better than a twin experiment. Also the comparison is not independent since MAR atmospheric variables are used as inputs of ORCHIDEE. The study therefore shows the difference of modelling approached in MAR and ORCHIDEE (in its various configurations). What is the objective of such comparison in regards with the objectives of the papers? Again, such question occurs because the context of the study is not stated loudly by the authors.
  Hopefully, the addition of PROMICE validation (see below) will help mitigate some of these concerns.

- I do realize it would be hard work to use independent data to validate the authors' approach but the absence of proper validation really weakens the validity of the approach and therefore the paper in its current version. One possibility could be to use data from GRACE about Greenland mass loss as a way to validate your approach by assuming all the mass loss is carried out by melting (in direct link with SMB). While this assumption is probably excessive as it ignores the impact of outlet glaciers, the comparison with independent data coming from GRACE could solve partially my aforementioned reservations about the validation aspects of this paper.
  We have highlighted the potential to use GRACE in the discussion. However, we do believe it is outside the scope of this paper since, as the reviewer acknowledges, this relies on a new set of assumptions and error treatment. Instead, we have chosen to validate with PROMICE, as mentioned in the following comment.

- Another possibility would be to use in situ data from the PROMICE network as mentioned in the conclusion L. 379-380 "One solution would be to run … lead to issues of scale and representativity". In land surface modelling, it is common for example to compare modelled soil moisture at e.g. 0.25° with in situ measurements, see e.g. Kumar

et al. (2019) or Albergel et al. (2020). While there are issues of scale, those comparisons are still by far important and very useful. I do not understand why it could not be done in the authors' context.

We agree that we were probably overly cautious in not including the evaluation against the in situ PROMICE data. This has now been added, as well as an introduction to PROMICE in the Methods and Data section, although we still caveat issues linked to scale representation. See below for the full text.

References:
- Albergel, C., Zheng, Y., Bonan, B., Dutra, E., Rodríguez-Fernández, N., Munier, S., Draper, C., de Rosnay, P., Muñoz-Sabater, J., Balsamo, G., Fairbairn, D., Meurey, C., and Calvet, J.-C.: Data assimilation for continuous global assessment of severe conditions over terrestrial surfaces, Hydrol. Earth Syst. Sci., 24, 4291–4316, https://doi.org/10.5194/hess-24-4291-2020, 2020.
- Kumar, S. V., Mocko, D. M., Wang, S., Peters-Lidard, C. D., and Borak, J.: Assimilation of remotely sensed Leaf Area Index into the Noah-MP land surface model: Impacts on water and carbon fluxes and states over the Continental U.S., J. Hydrometeorol., https://doi.org/10.1175/JHM-D-18-0237.1, 2019.

"3.3.2 Evaluation over PROMICE in situ sites

To evaluate the success of the optimisation, it is important to confront the results with different data. Here we look at how the fit against albedo at in situ sites is improved with the optimisation (Fig. 6). Generally, the albedo is found to improve - the fit to the observations results in a lower RMSD compared to when using the prior model. With the exception of UPE, reductions in RMSD are greater for the upper sites (between 11 and 25%) than for the lower sites (between -6 and 8%, where negative means the fit has degraded). For the UPE sites, this is the opposite. Of the 24 sites tested, the fit to the observations is only degraded in three cases. These sites are all lower sites - i.e., where the measurement station is near the ice sheet margin, where processes are harder to model. Two sites are found on the eastern edge of the ice sheet (SCO_L, TAS_L), and the last one is found at the southern tip of the ice sheet (QAS_L). When comparing to Fig. 4b, we can see that the eastern edge of the ice sheet is where the largest errors occur, even after the optimisation. Furthermore, TAS_L and QAS_L are two locations where the smallest amplitude and highest winter temperatures occur (van As et al., 2011, Fig.1) due to being exposed to the relatively warm wintertime atmospheric conditions of the Atlantic Ocean.

Figure 6 also shows us how ORCHIDEE generally performs at these sites - the magnitude of the RMSD remains similar for both parameter sets. Since the sites are mainly found at the edges of the ice sheet, errors are generally high - between 0.15 and 0.32. The two sites with the lowest RMSD for both the prior and posterior models are the ones located near the middle of the ice sheet, in the accumulation area (KAN_U and EGP). There is no obvious link between latitude and the magnitude of the errors.

Instead, elevation due to the position on the edges of the ice sheet is a more important factor.

Overall, this evaluation is encouraging - it shows that the optimisation was successful at improving model albedo when tested against a different data source. Nevertheless, we do need to highlight a couple of shortcomings in this comparison. Firstly, we do not have accurate local forcing data at the sites with which to drive ORCHIDEE. Therefore, the 20km MAR data was used, meaning that we are comparing observations and the model at different resolutions. Secondly, MODIS has been validated, and some of its biases due to the solar zenith angle were corrected for using PROMICE data (see Sect. 2.2.2). As such, the MODIS data used in the optimisation is not completely independent from the PROMICE data used in this evaluation.

[Figure]

**Fig 6.:** Evaluation of model-observation fit over PROMICE sites. For each year of available data, the RMSD for the months (Mar-Oct) is calculated. The mean over these RMSD values is shown in the figure. Points below the 1-to-1 line represent sites where the model-data fit is improved by the optimisation.

"

- If no independent data are used, then the part of comparing ORCHIDEE and MAR should be strengthen by highlighting the differences in term of modelling for both approaches (with appropriate bibliographical literature) and this comparison should be strongly linked with the Morris sensitivity analysis and with the background context of the study that is definitely missing. Also for SMB and runoff, the difference between MAR and ORCHIDEE clearly occur at the edges of the GrIS (either with standard, tuned or optimised parameters for ORCHIDEE). The parameter optimization for albedo does not make ORCHIDEE closer to MAR (quite the contrary). The previously performed Morris

sensitivity analysis could help to understand the mechanisms behind those increased differences and would make a nice discussion section.

A full comparison of both models, with additional literature and parallels to the SA, would be great. However, this would involve a lot more work in understanding the intricacies of both models. We do have a paper in preparation, Charbit et al., which will discuss a lot more the ORCHIDEE snow model. A more thorough comparison with MAR is undertaken in that study.

Nevertheless, we have added a couple of additional sentences to the MAR comparison:
> "*The fact that MAR has a more complex snow model that works better at capturing the different processes over Greenland leads us to believe structural changes are needed in ORCHIDEE for it to be able to better simulate SMB and its components. Through the optimisation, we have improved the representation of albedo but not of SMB and its components. This is because albedo is not the only important parameter in the modelling of the snowpack evolution. Other processes like melting depend on the snow's temperature profile, compaction, and refreezing, therefore on the thermal and mechanical properties of the snowpack. These processes must be well represented in the model and may require further calibration in future works.*"

MINOR COMMENTS AND TYPOS

L. 8 "This improvement is consistent for all years, even those not used in the calibration step". Could the authors rephrase the sentence as I would expect such result otherwise the methodology would not work?
We believe this statement is necessary since the model's structural errors could still prevent an improvement over evaluation years. This is especially true if different years suffer different conditions, such as increased atmospheric temperature and/or extreme events (e.g., volcanos depositing dust).

L. 14-16 "Increased warming … algae growth (Cook et al., 2020)". Darkening of GrIS has already observed and expect to worsen and increased impact on GrIS melting. Rephrase accordingly. Also, several missing references, among others:
- Dumont, M., Brun, E., Picard, G., Michou, M., Libois, Q., Petit, J.-R., Geyer, M., Morin, S. and Josse, B.: Contribution of light-absorbing impurities in snow to Greenland's darkening since 2009, Nature Geosci., 7, 509-512, 2014.
- Williamson, C. J., Cook, J., Tedstone, A., Yallop, M., McCutcheon, J., Poniecka, E., Campbell, D., Irvine-Fynn, T., McQuaid, J., Tranter, M., Perkins, R. and Anesio, A.: Algal photophysiology drives darkening and melt of the Greenland Ice Sheet, PNAS, 117(11), 5694-5705, 2020.
- Perini, L., Gostinčar, C., Anesio, A. M., Williamson, C., Tranter, M. and Gunde-Cimerman, N.: Darkening of the Greenland Ice Sheet: Fungal Abundance and Diversity Are Associated With Algal Bloom, Front. Microbiol., 10, https://doi.org/10.3389/fmicb.2019.00557, 2019.

Text rephrased and citations added.

L. 18-19 "This, in turn, enhances melting, creating feedback to the atmosphere". Missing reference to support the statement: •
- Le clec'h, S., Charbit, S., Quiquet, A., Fettweis, X., Dumas, C., Kageyama, M., Wyard, C., and Ritz, C.: Assessment of the Greenland ice sheet–atmosphere feedbacks for the next century with a regional atmospheric model coupled to an ice sheet model, The Cryosphere, 13, 373–395, https://doi.org/10.5194/tc-13-373-2019, 2019.
- Box, J. E., Werhlé, A., van As, D., Fausto, R. S., Kjeldsen, K. K., Dachauer, A., Alhstrøm, A. P. and Picard, G.: Greenland Ice Sheet Rainfall, Heat and Albedo Feedback Impacts From the Mid-August 2021 Atmospheric River, Geophys. Res. Lett., https://doi.org/10.1029/2021GL097356, 2022.

Citations added

L. 22: "… it is crucial that it is accurately simulated in THE land surface models (LSMs) …"
Corrected

L. 39-40: "Examples of DA used for parameter estimation … in snow modelling are less common. Bonan et al. (2014) … ". The reference is not about snow modelling but on ice sheet initialization and DA or inverse modelling is well known in this field, see the following paragraph in Bonan et al. (2014) "MacAyeal (1992) and MacAyeal (1993) introduced control methods to infer basal drag in ice-stream models, using in particular the self-adjoint property of such models, leading to many application papers (Rommelaere and MacAyeal, 1997; Vieli and Payne, 2003), and later for full Stokes models (Morlighem et al., 2010; Jay-Allemand et al., 2011). Later on, many DA and inverse methods were introduced in glaciology. The Best Linear Unbiased Estimation (BLUE) and Optimal Interpolation (OI) methods were introduced by Arthern (2003) and Berliner et al. (2008). The Robin inverse method due to Chaabane and Jaoua (1999) has been introduced by Arthern and Gudmundsson (2010) for ice sheet models, and finally Heimbach and Bugnion (2009) presented the first adjoint ice sheet model derived automatically." Most previous references focuses on estimating basal friction or basal velocities as parameters. Regarding parameter estimation for ice sheet mass balance, you can see:
- Bonan, B., Nodet. M., Ozenda, O. and Ritz, C.: Data assimilation in glaciology, in Advanced Data Assimilation for Geosciences, Lecture Notes of the Les Houches School of Physics: Special Issue June 2012 (Edited by Blayo, E., Bocquet, M., Cosme, E. and Cugliandolo, L. M.), 577-584, Oxford University Press, 2014.

We have changed the reference to that of Su et al., 2011 which discusses joint parameter and state estimation in snow modelling.

Reference:
> Su, Hua, et al. "Parameter estimation in ensemble based snow data assimilation: A synthetic study." *Advances in water resources* 34.3 (2011): 407-416.

L. 48-49: "However, with large amounts of data, … the multisite approach is common":
Debatable statement. One of the main reason of the multisite approach is commonly used is

that the set of optimal parameters or parametrizations for various sites in LSMs can differ significantly due to soil properties (soil texture, water potential, hydraulic conductivity …) and land cover (vegetation variables). Please soften the previous sentence accordingly.
Removed the part "multisite approach is common" to soften the statement.

L. 64: authors indicate that CMIP 6 version of ORCHIDEE is used in this paper: add reference publication(s) for this version in addition to the historic paper of Krinner et al. (2005). Also authors mention in the code availability section that ORCHIDEE vAR6 is employed. Could the authors harmonize notations between both paragraphs?
The citation Krinner et al., (2015) has been added to this section.The citations Boucher et al., 2020 and Cheruy et al., (2020) can be found on L61. These described the IPSL ESM of which ORCHIDEE is the terrestrial component. Unfortunately, there are no other more recent publications describing ORCHIDEE. We have expanded the code availability section to explain that vAR6 is the version used for the CMIP6 exercice.

L. 73-74 "we computed the mean of albedo in both visible (VIS) and near-infrared (NIR) spectral domains". Please indicate that this is to be in accordance with MODIS data. Also the description of albedo following this sentence does not distinguish VIS and NIR spectral domains. I do not think spectral domains intervene in the computation of modelled albedo. Rephrase sentence L. 73-74 in order to state that your model does not distinguish VIS and NIR albedo.
We have added the following: *"This is done to be in accordance with MODIS data".* ORCHIDEE does calculate both VIS and NIR albedo which is why taking the mean was necessary.

Section 2.2: Could the authors cite the paper(s) associated with MAR Version 3.11.4? Gallée and Schayes (1994) is rather outdated for this version.
The following citation has been added:
Kittel, C.: Present and future sensitivity of the Antarctic surface mass balance to oceanic and atmospheric forcings: insights with the regional climate model MAR, PhD Thesis, Université de Liège, Liège, Belgique, Liège, 2021.

Section 2.3: Please mention that this dataset do not include data from the Aqua satellite as explained in Box et al. (2017). People familiar with MODIS datasets tend to expect data coming from both Terra and Aqua satellites.
Added to L101:
"Note that this dataset does not include data from the Aqua satellite."

L. 106-108 "Finally, in this dataset, … in the April values". This statement is rather confusing, please rephrase.
Rephrased as follows:
"*Finally, in this dataset, the April values are used for the winter months (January, February, November, and December). This is because there is inadequate solar illumination to compute the albedo during these months.*"

L. 114-120 "Bayesian statistical formalism (Tarantola, 2005)". The formulation of the cost function can also be seen as an optimal control problem without any assumption on probabilistic distributions. This has given the basis of 3D and 4D-Var approach, see for example Nichols (2010). Could the authors rephrase L.114 to L.120 to play down the emphasis on the Bayesian statistical formalism?

- Reference: Nichols, N. K.: Mathematical concepts of data assimilation, in: Data assimilation: making sense of observations, edited by: Lahoz, W., Khattatov, B. and Menard, R., Springer-Verlag, Berlin, Germany, 13–40, 2010

Thank you for forwarding the Nichols' paper - it was a very interesting read. This paper describes data assimilation mainly for the correction of model state. The problem of state estimation can effectively be seen as a control problem and it is true that the cost function we define can be used in this different context. Nevertheless, we believe that using data assimilation for parameter estimation is very different. We have comparatively few parameters and these usually represent biogeophysical properties that we can sometimes measure and for which we can set ranges. We, therefore, believe it is essential to add prior information to the assimilation which makes the problem well adapted to Bayes' theorem. Furthermore, the literature in the field of land-surface modelling parameter estimation uses this terminology; from all the studies from our group (https://orchidas.lsce.ipsl.fr/publications.php) to the early BETHY optimisation (e.g., Knorr and Kattge, 2005). As such, we do not feel comfortable removing the Bayesian term from the ORCHIDAS description. However, we have downplayed it in the rest of the text (see responses below).

Reference
Knorr, Wolfgang, and Jens Kattge. "Inversion of terrestrial ecosystem model parameter values against eddy covariance measurements by Monte Carlo sampling." *Global change biology* 11.8 (2005): 1333-1351.

L. 163 "they correspond to ablation areas" Most parts of Greenland nowadays experience ablation during summer. Edges are where strong ablation occurs. Please rephrase accordingly.
Changed to: *"to areas of strong ablation"*

L. 168 "… into THE ORCHIDEE, where … "
Done

L. 174-176 "They were also the pixels with the largest errors when compared … with RMSD greater than 0.1". When this calculation is performed? Before calibration or after? Can the author explain where does this number comes from? It can be simply done by referring to a subsequent section of the paper.
This is the error in model prior to calibration - this has been clarified in the text. This calculation was done when testing whether the threshold of 0.5 would be satisfactory for defining edges (not shown in the paper).

L. 181 "… to allow THE snow to accumulate …"
Removed

L. 209 "Bayesian framework" see previous comment on the Bayesian term
Replaced with minimisation algorithm

About Figure 2: "currently operational ORCHIDEE version" By currently operational ORCHIDEE
version, did the author mean ORCHIDEE with parameters set at default values (as in Table
B1)? To my knowledge, the way albedo is modelled and the new "icy" type cannot be called
"operational" yet. Could the authors modify the legend of Figure2 to reflect this point?
Changed to "standard ORCHIDEE version (before tuning)"

L. 223 "… affect by THE albedo parameters …"
Removed

L. 297 "Bayesian framework" see previous comment on the Bayesian term
"Bayesian" removed

L. 302 I think the text should refer to Figure 4a instead of Figure 6a here.
Changed

L. 308 Replace "omega" by ω and "beta" by β. L. 320 "… different parameter setS on modelling
…"
Done

L. 326-327 "Compared to MAR, the manually … ORCHIDEE performs best at simulating SMB".
Best performance does not really make sense in the context of comparing two models. This
sentence and the rest of the section should be rewritten keeping this fact in mind.
Sentence rewritten as follows:
    "*The manually tuned version of ORCHIDEE simulates SMB the closest to MAR's SMB*"

L. 347 "Bayesian optimisation" see previous comment on the Bayesian term
"Bayesian" removed from the line

L. 347-348 "However, we overfitted to albedo with no other data". This statement raises the
question of prescribed observation error variances. Would other prescribed values have made
the impact of optimized parameters for ORCHIDEE more in line with MAR? Or is this question
related to modelling differences? Could the authors reflect on that question in the paragraph?
Even though we have prescribed relatively large error variances, we are still able to fit the
MODIS retrievals reasonably well. We do not think other prescribed values would have made
ORCHIDEE more in line with MAR, since there are large modelling differences between MAR
and ORCHIDEE.  MAR has a much more complex snow model. For example, it works over a
number of adjustable layers (generally comprised between 30 and 50)  compared to
ORCHIDEE's three. Furthermore, albedo is not the only factor impacting SMB. We fit against
albedo in the optimisation but do not include any data for runoff, or sublimation for example.
This paragraph has been expanded as follows:

*"The fact that MAR has a more complex snow model that works better at capturing the different processes over Greenland leads us to believe structural changes are needed in ORCHIDEE for it to be able to better simulate SMB and its components. Through the optimisation, we have improved the representation of albedo but not of SMB and its components. This is because albedo is not the only important parameter in the modelling of the snowpack evolution. Other processes like melting depend on the snow's temperature profile, compaction, and refreezing, therefore on the thermal and mechanical properties of the snowpack. These processes must be well represented in the model and may require further calibration in future works."*

L 360-361 "When we cannot further improve … this can point to structural deficiencies in the model". I tend to disagree with this statement. Parameter estimations can sometimes hide structural model deficiencies, i.e. you may obtain the right results but for the wrong reason. Could the authors weight on that comment?

It is true that parameter estimation can hide structural model deficiencies. It is also true that if we cannot fit the data, it is likely that something is wrong in the model for example a missing process. The two are not mutually exclusive. We have added the following to L360:

"... it can help identify structural issues in the model - *although we do need to be cautious since parameter estimation can also hide structural model deficiencies*.

L. 361-362 "For example, we cannot capture the different albedos in the north and the south of the ice sheet with the current processes represented" This problem might also come from that the author assume the same set of parameters for the whole Greenland. A multisite approach may have reduce this problem (perhaps for the wrong reasons). Could the authors reflect on that question?

As stated in the reply above, we want to have one set of parameters for the whole of Greenland. If we start having a lot of different parameter sets for different regions, this becomes hard to deal with and to justify when to stop having more cases. Furthermore, since the climate over Greenland is expected to change, so parameters optimised in the south, where it is currently warmer, may be valid later in the north as temperatures change. By having one set of parameters, we are more likely to have averaged out over these different conditions having the set more robust.

L. 375 "There is an urgent need for data producers to provide this uncertainty, ideally at each time step". I could not agree more. The authors can mention this following reference to strengthen their statement:

- Merchant, C. J., Paul, F., Popp, T., Ablain, M., Bontemps, S., Defourny, P., Hollmann, R., Lavergne, T., Laeng, A., de Leeuw, G., Mittaz, J., Poulsen, C., Povey, A. C., Reuter, M., Sathyendranath, S., Sandven, S., Sofieva, V. F., and Wagner, W.: Uncertainty information in climate data records from Earth observation, Earth Syst. Sci. Data, 9, 511–527, https://doi.org/10.5194/essd-9-511-2017, 2017.

Thank you for the citation; this has been added to the text

L. 379-380 "One solution would be to run … lead to issues of scale and representativity REPRESENTATIVENESS".

Corrected

L. 395 Rephrase the sentence to replace "better" by "more consistent with MAR outputs"

Done

L. 398-404 I have some reservations with the statements written in the paragraph. Would the idea behind using all these satellite datasets be to replace the modelling of the Greenland ice sheet ice dynamics? Could the authors temper those statements?

These data would be used to optimise the internal parameters of the model, further ensuring that different processes were constrained, not just albedo. To clarify this, we added "*model's internal parameters*" to L398 and L404.

---

## Author Response (AR3)

**Report #1**

Dear authors,

first I would like to acknowledge the impressive work you did to include all my remarks, especially on validating your approach with PROMICE in-situ measurements and on clarifying the purposes of your study. I believe that, in its current form, this paper forms a valuable contribution on albedo modelling and assimilation over Greenland. Therefore, and with great pleasure, I recommend the manuscript to be accepted subject to the following corrections/precisions:

We would like to thank the reviewer for their appreciation of our efforts and their recommendation.

Fig. 8 shows a negative sublimation for every version of ORCHIDEE during winter time. Does a negative sublimation make sense? If so, could the authors tell us more about that subject? If this does not make sense, this may have an impact on all the study linked to sublimation. Could the authors double-check what they consider as sublimation?

Negative values for latent heat flux and therefore sublimation are indeed observed. These are linked to colder winter surface temperatures in the ORCHIDEE compared to the MAR model. We have added the following text:

> "and no sublimation is simulated by ORCHIDEE outside the summer months. **Indeed in winter, we even get negative values, i.e., condensation. This is likely due to the fact that surface temperatures are generally lower than those from MAR leading to a lower saturation humidity and thus to condensation.**"

L. 41: ... albedo data was also used by Wang et al. (2015) to calibrate ... [no need to put Wang et al. in parenthesis]

Corrected

L. 43: Crocus instead of Corcus

Corrected

L. 149- 150: Please rephrase "See Fausto et al (2021) ..." such as "Further information on ground measurements of snow albedo and associated methodology can be found in Fausto et al. (2021)".

Done

**Report #2**

I appreciate all of the work that the authors have done to clarify the text and improve the figures. In particular, I appreciate the care taken in the new colour choices.
I think that the communication of the results has been improved. I have a few minor comments to make about this version, but I would accept the manuscript
after these suggestions are addressed.

We would like to thank the reviewer for their taking the time to reread the manuscripts and for the final comments.

In 3 places the sub-subsection numbers are not incremented properly. Please recheck the section titles to ensure proper numbering.

"Impact of different parameter sets" is now under the "Results" section.

Introduction:
L7 I would consider writing out the acronym RMSD.

Expanded

L11 I would explicitly say what you are optimising: "...for optimisation." Is this for optimisation of all a model's parameters or specifically for the albedo?
Changed to "when optimisation these parameters"

Body:
L44 Crocus is misspelled as corcus
Corrected
L74 You already have defined LSM and DA so use the acronyms
Changed
L83 Earth system model should be Earth System Model
Changed
L97 Does "non-ice-free regions" mean"ice-covered land"? non-ice-free is a confusing way to put it, in my opinion.
Agreed we have corrected to ice-free
L109 What is "the latter term of f_age"? Is it the exponential in the second term of the numerator?
Missing comma when added - this statement refers to the previous equation, where fage is the latter term
Table 1 Is tau_max given in days? I think so, but need to add the units, please.
Unit added

L136 "further processed by Box" should be more like: "further processed by applying the techniques proposed by Box"
This line describes the processing done by Box et al - since we do not apply the technique but simply use their processed version of albedo we prefer to leave the text as it is.

L144 I think that the the dataset should be clarified. Is it the snow albedo dataset created by Box? I'm not sure.
Changed to "this dataset processed by Box et al."

L149, can use the acronym "GRIS"
Changed

L165 How was the Background error covariance matrix created? I don't mean to suggest that you write a new section or anything. It is just that B is very important, so some words to explain how it is defined would be helpful.
Added "For the B matrix, we define the prior distribution of each parameter to be 40% of the prior range"

L222 exceeds and not exceed.
Corrected
L230 "Performed Experiments" could just read "Experiments"
Changed

Figure 2 I suggest that the panels have labels like a, b, c and d. Then the caption will be easier to write. Then there is no need to specify reading the figure from left to right.
Added

L399-404. I am happy to see this paragraph included in the revised version. As a reader, it clarified what should be done next in this kind of study.
Thank you

L450. I would change "we get the most different results" to something like "the differences between SMB as determined by MAR and by ORCHIDEE are larger"
This paragraph is looking at sublimation. We have rephrased as follows: "the differences between each ORCHIDEE simulation are most marked"

L551 Should be "Further work will include"
Corrected
L558 Perhaps clearer to re-write as " discretising the snowpack vertically"

Done

L564 Perhaps be more explicit here and write: "the retrievals of the albedo from the observed quantity".

Changed

L594. This is not clear to me at all: "to distinguish nonlinearity from interactions" An interaction can be nonlinear. Do you mean to distinguish nonlinear interactions from linear interactions? What I mean is that nonlinearity is not clearer defined in opposition to interactions. Please could you clarify?

The input parameters can have linear or non-linear effects on the output. This can be can not be distinguished from the effect the parameters will have on each other. This has been clarified in the text: "it is not possible to distinguish the nonlinear effect individual parameters have on the model output from the effect of their interactions with other parameters"

Appendix B1 has a title "Parameter Values" but there is no text in this section. Should there be a single section Appendix B?

Section B1 contains Table B1. We have added the following text: "In Table B1, we list the different parameter values used and found in this study."